# S100A8/A9 as a prognostic biomarker with causal effects for post-acute myocardial infarction heart failure

Jie Ma[1,2,5], Yang Li[1,2,5], Ping Li[1,5], Xinying Yang[1,2], Shuolin Zhu[1,2], Ke Ma[1,2], Fei Gao[1], Hai Gao[1], Hui Zhang[3], Xin-liang Ma [4], Jie Du[1,2] & Yulin Li [1,2] ✉

Heart failure is the prevalent complication of acute myocardial infarction. We aim to identify a biomarker for heart failure post-acute myocardial infarction. This observational study includes 1062 and 1043 patients with acute myocardial infarction in the discovery and validation cohorts, respectively. The outcomes are in-hospital and long-term heart failure events. S100A8/A9 is screened out through proteomic analysis, and elevated circulating S100A8/A9 is independently associated with heart failure in discovery and validation cohorts. Furthermore, the predictive value of S100A8/A9 is superior to the traditional biomarkers, and the addition of S100A8/A9 improves the risk estimation using traditional risk factors. We finally report causal effect of S100A8/A9 on heart failure in three independent cohorts using Mendelian randomization approach. Here, we show that S100A8/A9 is a predictor and potentially causal medicator for heart failure post-acute myocardial infarction.

Despite therapeutic advancements, patients with acute myocardial infarction (AMI) exhibit a high risk of adverse cardiovascular outcomes[1]. Heart failure (HF), a common complication following the first AMI[2], is strongly associated with reduced in-hospital and long-term survival[3]. Current guidelines recommend that patients undergo early post-AMI risk assessment for appropriate therapy provision. Early and accurate risk stratification to identify patients with impending HF is crucial to guide treatment and improve prognosis[4,5]. Clinical risk scores, including the Thrombolysis in Myocardial Infarction and Global Registry of Acute Coronary Events scores, can only predict recurrent myocardial infarction (MI) and death[6,7]. Existing clinical biomarkers, including myocardial necrosis (cardiac troponin I, cTnI), stress (B-type natriuretic peptide, BNP), and inflammation (high-sensitivity C-reactive protein, hs-CRP), are insufficient for precise HF prediction in patients with AMI[5]. Most existing biomarkers only demonstrate correlation, rather than a causal relationship, with HF, and may emerge because of confounders or reverse causation, allowing them to act as passive bystanders rather than drivers of HF. If a biomarker is causally associated with HF, it helps predict HF, and serves as the target of intervention.

Unlike traditional hypothesis-driven approaches, unbiased and high-throughput strategies enable the discovery of novel biomarkers with improved predictions and novel mechanisms that provide insights into disease development. Proteomic profiles represent sources of new candidate biomarkers with diagnostic and prognostic value[8]. Mendelian randomization (MR) study is used to infer causality from observational data because germline genetic variations are defined at conception and are generally not associated with conventional confounders in observational studies[9]. If genetically predicted portion of biomarker is associated with the outcome and meets several strict assumptions, the measured marker might has a causal effect on outcomes.

In this study, we aim to identify biomarkers associated with HF development in patients with AMI. We select S100A8/A9 as a potential biomarker using proteomic analyses. S100A8 and S100A9, as endogenous alarmins, are constitutively expressed in myeloid cells and stored as granules that are ready to be released in response to infectious

[1]Beijing Anzhen Hospital of Capital Medical University, Beijing, China. [2]Beijing Institute of Heart Lung and Blood Vessel Diseases, Beijing, China. [3]Department of Preventive Medicine, Feinberg School of Medicine, Northwestern University, Chicago, IL, USA. [4]Department of Emergency Medicine, Thomas Jefferson University, Philadelphia, PA, USA. [5]These authors contributed equally: Jie Ma, Yang Li, Ping Li. ✉e-mail: lyllyl_1111@163.com

or bacteria-free inflammation. They exist in several forms but preferentially form a heterodimeric complex of S100A8/A9, which is necessary for their biological effects[10]. The secretion of S100A8/A9 is partly dependent on the reactive oxygen species (ROS) and potassium Efflux. S100A8/A9 is also released during NETosis. In the MI setting, both excessive ROS and NETosis are conductive to S100A8/A9 release into the heart and circulation[11]. Consequently, we prospectively validate the predictive values of S100A8/A9 in two independent cohorts and evaluate the causal relationship between S100A8/A9 and post-AMI HF.

## Results

### Patient characteristics

The study design is shown in Fig. 1 and includes three steps. The clinicopathological characteristics of patients with AMI ($n = 20$) and healthy controls (HCs) ($n = 10$) in step 1 are presented in Supplementary Table 1. Baseline information of the discovery ($n = 1062$) and validation ($n = 1043$) cohorts in steps 2 and 3 was displayed by HF status (Table 1, Supplementary Fig. 1). The proportions of ST-segment myocardial infraction (STEMI) vs. non-ST-segment myocardial infraction (NSTEMI) were 75% vs. 25% in discovery cohort and 61.6% vs. 38.4% in validation cohort. During follow-up, 118 in-hospital (11%) and 178 long-term (17%) HF events were recorded in the discovery cohort, and 110 in-hospital (11%) and 82 long-term (8%) HF events were recorded in the validation cohort (Supplementary Tables 2 and 3). In the discovery cohort, patients with HF were older; had a higher proportion of Killip classification III; higher serum creatinine, fasting glucose, neutrophil counts, and biomarkers (cTnI, BNP, hs-CRP) levels; lower blood pressure (systolic and diastolic blood pressure) and left ventricular ejection fraction (LVEF); larger infarct size; and more left main lesions. Medications and multivessel disease were similar between patients with and without HF. In the validation cohort, patients with AMI and HF had lower blood pressure, larger infarct size, and higher creatinine, neutrophil counts, and biomarker levels. The incidence of HF events was higher in the discovery cohort than in the validation cohort (28% vs. 18%) owing to the longer follow-up period.

Some differences in HF risk characteristics can be observed among patients in the validation cohort compared to those in the discovery cohort (Supplementary Table 4). For example, the validation cohort had more younger and male patients, higher systolic blood pressure at admission, and fewer patients with a history of comorbidities than the discovery cohort. Several laboratory indicators, including glucose, lipid, creatinine, BNP, cTnI, and CRP levels, were also significantly lower in the validation cohort than in the discovery cohort.

The baseline information of the general population ($n = 588$) in step 3 is presented in Supplementary Table 5.

### Screening of prognostic biomarkers

For non-biased screening of HF-related proteins, we quantified 1000 proteins in serum samples from 20 patients with AMI (HF, $n = 10$; no-HF, $n = 10$) and ten HCs at admission (Supplementary Fig. 2, Supplementary Data 1). Differential expression analysis identified 362 differentially expressed proteins (DEPs) in patients with AMI vs. HCs, 142 DEPs in patients without HF vs. those with HF development, and 134 overlapping HF-related DEPs between the two comparisons. Among them, 108 proteins had higher levels, while the remaining 26 proteins had lower levels in patients with HF than in HCs or patients with AMI without HF (Supplementary Fig. 3a). The 134 HF-related proteins included those with established prognostic values for HF (BNP, CRP, cTnT, and several emerging candidates) (Supplementary Table 6). S100A12/A8/A9 were the top-ranked DEPs by either fold change or statistical significance in the HF vs. no-HF groups (Supplementary Table 6). Subsequently, in the least absolute shrinkage and selection operator regression analyses based on 134 HF-related proteins, the best prognostic candidates were S100A8/A9 and S100A12 (Supplementary Fig. 4). We investigated the potential biological functions of the HF-related proteins. In the Reactome enrichment analysis, HF-related proteins were primarily enriched in inflammatory pathways (false discovery rate < 0.05), including toll-like receptor (TLR) and interleukin signaling (Supplementary Fig. 3b), suggesting an inflammatory activation signature in patients progressing to HF. Collectively, the serum proteomics suggests that S100A8/A9 and S100A12 are the best prognostic candidates for HF in patients with AMI.

### Predictive values of S100A8/A9 and S100A12 for HF in discovery cohort

To examine the association between the candidate proteins and HF, we measured circulating S100A8/A9 and S100A12 levels at admission in

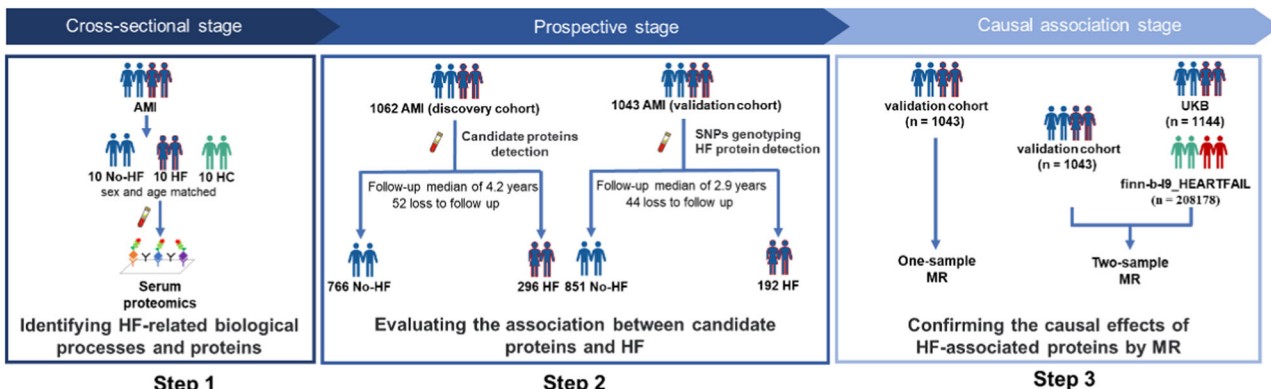

**Fig. 1 | Study design.** This study comprised the following three steps: (1) HF-related biological processes and proteins were identified using serum proteomics in a cross-sectional set of patients with AMI who developed HF during hospitalization ($n = 10$), patients with AMI without HF ($n = 10$), and HCs ($n = 10$). (2) The association between candidate proteins and HF was prospectively evaluated in the discovery (HF: $n = 296$, no-HF: $n = 766$) and validation (HF: $n = 192$, no-HF: $n = 851$) cohorts. (3) The causal relationship between HF-associated proteins and HF was confirmed using MR analysis. For individual-level one-sample MR analysis, the causal association of genetic instruments with post-AMI HF events was assessed in the validation cohort (HF: $n = 192$, no-HF: $n = 851$). For the two-sample MR analysis, estimates of the association between the genetic instruments and S100A8/A9 levels from the validation cohort ($n = 1043$) and the association between the genetic instruments and post-AMI HF from the UKB cohort ($n = 1144$) were used to examine the causal effect of S100A8/A9 levels on post-AMI HF. Moreover, a statistical summary of the association between genetic instruments and S100A9 levels from the validation cohort and a statistical summary of the GWAS from the finn-b-I9_HEARTFAIL study ($n = 208178$) were used to evaluate the causal association between genetic instruments and general HF. AMI acute myocardial infarction, HC healthy control, HF heart failure, MR Mendelian randomization, UKB UK Biobank.

**Table 1 | Clinical characteristics according to HF events in the discovery and validation cohorts**

| Variables | Discovery cohort (n = 1062) | | | Validation cohort (n = 1043) | | |
|---|---|---|---|---|---|---|
| | HF events (n = 296) | No-HF events (n = 766) | P-value | HF events (n = 192) | No-HF events (n = 851) | P-value |
| **Demographics** | | | | | | |
| Age (years) | 60.5 (53.0–70.0) | 59.0 (50.0–66.0) | **0.001** | 40.0 (37.0–42.0) | 39.0 (36.0–42.0) | 0.110 |
| Male sex | 225 (76.0) | 626 (81.7) | **0.037** | 179 (93.2) | 772 (90.7) | 0.268 |
| SBP (mm Hg) | 116.0 (103.3–134.0) | 120.0 (110.0–136.0) | **4.03E-05** | 120.0 (110.0–130.0) | 125.0 (119.0–134.0) | **7.76E-06** |
| DBP (mm Hg) | 70.0 (63.3–80.0) | 74.0 (70.0–82.0) | **0.002** | 75.0 (70.0–80.0) | 80.0 (70.0–85.0) | **0.004** |
| Current smoking | 187 (63.2) | 465 (60.7) | 0.458 | 130 (67.7) | 550 (64.6) | 0.419 |
| **Killip classification** | | | | | | |
| I | 230 (77.7) | 649 (84.7) | **0.007** | 145(75.5) | 826(97.1) | **2.04E-26** |
| II | 53 (17.9) | 110 (14.4) | 0.151 | 33(17.2) | 25(2.9) | **7.10E-15** |
| III | 13 (4.4) | 7 (0.9) | **1.85E-04** | 14(7.3) | 0(0.0) | **3.45E-11** |
| **Medical history** | | | | | | |
| Hypertension | 177 (59.8) | 444 (58.0) | 0.587 | 102 (53.1) | 411(48.3) | 0.227 |
| Hyperlipidemia | 197 (66.6) | 498 (65.0) | 0.636 | 93 (48.4) | 389(45.7) | 0.494 |
| Diabetes mellitus | 106 (35.8) | 249 (32.5) | 0.306 | 47 (24.5) | 166 (19.5) | 0.123 |
| CAD | 90 (30.4) | 229 (29.9) | 0.871 | 41(21.4) | 168(19.7) | 0.614 |
| **Biochemical** | | | | | | |
| Neutrophil counts (×10⁹/L) | 6.8 (5.3–9.1) | 5.9 (4.5–8.1) | **8.21E-07** | 6.8 (5.0–8.9) | 5.7 (4.5–7.5) | **3.87E-06** |
| HDL cholesterol (mmol/L) | 1.0 (0.9–1.2)[a] | 1.0 (0.9–1.2)[a] | 0.913 | 0.9 (0.8–1.1) | 1.0 (0.8–1.1) | 0.163 |
| LDL cholesterol (mmol/L) | 2.9 (2.2–3.5)[a] | 2.8 (2.3–3.5)[a] | 0.647 | 2.3 (1.7–3.1) | 2.3 (1.7–3.0) | 0.907 |
| Fasting glucose (mmol/L) | 7.1 (5.8–10.0) | 6.6 (5.5–8.9) | **0.001** | 5.5 (5.0–7.2) | 5.5 (5.0–6.4) | 0.379 |
| Creatinine (µmol/L) | 76.4 (64.7–87.6) | 71.9 (61.6–83.2) | **0.001** | 73.9 (65.4–85.4) | 70.4 (63.6–78.4) | **8.42E-05** |
| **Biomarkers** | | | | | | |
| cTnI (ng/mL) | 2.5 (0.2–16.9) | 1.3 (0.1–7.8) | **0.005** | 2.1 (0.3–19.8) | 0.8 (0.2–4.6) | **1.70E-05** |
| BNP (pg/mL) | 147.5 (57.8–360.0) | 97.0 (39.0–211.0) | **1.93E-07** | 149.9 (60.0–343.8) | 68.0 (27.0–168.3) | **3.08E-13** |
| hs-CRP (mg/L) | 7.9 (2.6–23.4) | 5.3 (1.9–17.0) | **2.14E-04** | 6.1 (2.2–16.7) | 4.4 (1.7–12.4) | **0.021** |
| **Overall lesion profiles** | | | | | | |
| Left main artery disease | 15 (5.1) | 17 (2.2) | **0.015** | 7 (3.6) | 33 (3.9) | 0.880 |
| 2-vessel disease | 86 (29.1) | 225 (29.4) | 0.918 | 30 (15.6) | 162 (19.0) | 0.271 |
| 3-vessel disease | 43 (14.5) | 94 (12.3) | 0.326 | 47 (24.5) | 179 (21.0) | 0.295 |
| **Echocardiography** | | | | | | |
| Admission LVEF (%) | 50.0 (45.0–56.0) | 55.0 (50.0–59.0) | **1.14E-12** | 50.0 (45.0–55.8) | 55.0 (51.0,60.0) | **2.61E-17** |
| **Infarct size** | | | | | | |
| CK-MB-based estimation(ng*h/ml) | 5648.9 (2354.4–7685.7) | 4825.4(1896.1–6944.4) | **0.001** | 4803.5 (3174.5–9716.8) | 4326.0 (2120.4–5864.0) | **3.95E-07** |
| **Medication at discharge** | | | | | | |
| Aspirin | 276 (93.2) | 762 (99.5) | 0.392 | 183 (95.3) | 827 (97.2) | 0.182 |
| P2Y12 receptor Inhibitor | 279 (94.3) | 766 (100.0) | \ | 175 (91.1) | 789 (92.7) | 0.458 |
| Statin | 272 (91.9) | 747 (97.5) | 0.979 | 180 (93.8) | 810 (95.2) | 0.414 |
| ACEI or ARB | 161 (54.4) | 445 (58.1) | 0.911 | 90 (46.9) | 359 (42.2) | 0.236 |
| Beta-blockers | 218 (73.6) | 565 (73.8) | 0.149 | 138 (71.9) | 591 (69.4) | 0.508 |
| MRA | 42 (14.2) | 133 (17.4) | 0.376 | 15 (7.8) | 60 (7.1) | 0.712 |

Data were presented as absolute numbers (percentages) or medians (interquartile ranges). Categorical variables were analyzed using a two-sided χ² or Fisher's exact test. Continuous parametric data were analyzed using a two-sided *t*-test. Continuous non-parametric data were analyzed using a two-sided the Mann–Whitney *U*-test. *P* < 0.05 was considered as significant. Source data are provided in the Source Data File. Bold indicated *P* < 0.05.

*ACEI* angiotensin-converting enzyme inhibitor, *ARB* angiotensin-receptor blocker, *BNP* B-type natriuretic peptide, *CAD* coronary artery disease, *cTnI* cardiac troponin I, *DBP* diastolic blood pressure, *HDL* high-density lipoprotein, *hs-CRP* high-sensitivity C-reactive protein, *LDL* low-density lipoprotein, *LVEF* left ventricular ejection fraction, *MRA* mineralocorticoid receptor antagonist, *P2Y12* receptor Inhibitor included clopidogrel and ticagrelor, *SBP* systolic blood pressure.

[a]1–4% data missing.

the discovery cohort. S100A8/A9 and S100A12 were increased in patients with HF compared to patients without HF. The levels of S100A8/A9 and S100A12 were increased by 1.5-fold and 1.1-fold in patients with HF compared to those without HF, respectively (Fig. 2a and Supplementary Fig. 5a). In univariable analysis, the hazard ratio (HR) for post-AMI HF per standard deviation (SD) of S100A8/A9 was 1.92 (1.69–2.19), *P* < 0.001, and that of S100A12 was 1.09 (0.97–1.24), *P* = 0.157 (Fig. 2b, Supplementary Table 7). In multivariate analysis, the

association between S100A8/A9 and HF remained significant after adjustment for sex and significant clinical variables chosen from the univariable analysis (*P* < 0.05) (HR per SD: model 1: 2.05 [95% confidence interval [CI]: 1.80–2.35], *P* < 0.001; model 2: 2.15 [95% CI: 1.88–2.46], *P* < 0.001; model 3: 2.03 [95% CI: 1.77–2.33], *P* < 0.001) (Fig. 2b). The associated between S100A12 and HF was not significant after adjustment (HR per SD for model 3: 1.08 [95% CI: 0.95–1.22], *P* = 0.261) (Supplementary Fig. 5b). Spline regression adjusted for the

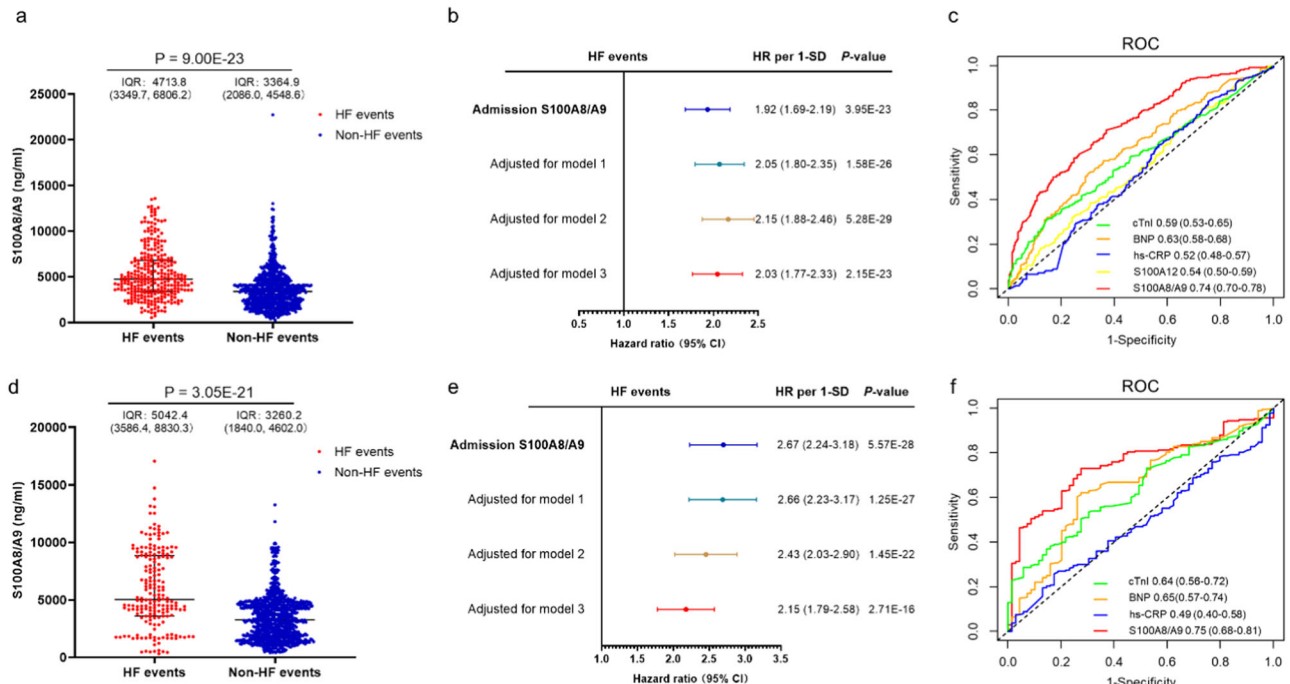

**Fig. 2 | Predictive value of S100A8/A9 for HF in discovery and validation cohort.** S100A8/A9 levels distribution at admission in the HF events (n = 296) and non-HF events (n = 766) groups of discovery cohort (**a**), as well as in the HF events (n = 192) and non-HF events (n = 851) groups of validation cohort (**d**). Red, patients who experienced HF events; blue, patients who did not experience HF events. The scatter plots in this figure show the median (center line), 25th, and 75th percentile (lower and upper boundary). Differences between the two groups were analyzed using a two-sided Wilcoxon-test. P < 0.05 was considered as significant. Unadjusted and adjusted HRs for HF events from Cox proportional hazards regression analysis were shown for discovery (n = 1062) and validation (n = 1043) cohort (**b** and **e**), which are indicated with the points and the bars showing the 95% confidence interval. S100A8/A9 was measured on admission. Model 1: adjusted for age and sex; model 2: adjusted for model 1+systolic blood pressure, Killip classification at admission, fasting glucose, creatinine, left main artery disease; model 3: adjusted for model 2+neutrophil count, cTnI, BNP, hs-CRP, left ventricular ejection fraction at admission and estimated infarct size (CK-MB AUC0-72). P-values reported are two-tailed from COX proportional-hazard regression analyses. P < 0.05 was considered as significant. ROC curves for biomarkers levels at admission in the discovery (**c**) and validation (**f**) cohort for HF events. Blue, hs-CRP; green, cTnI; orange, BNP; yellow, S100A12; red, S100A8/A9. HF heart failure, HRs hazard ratios, ROC receiver operating characteristics. Source data are provided as a Source Data file.

variables in model 3 showed a positive linear dose-response association between S100A8/A9 and HF (Supplementary Fig. 6). Moreover, a significant competing risk of events was observed in post-AMI populations. For instance, death not caused by HF was a competing risk factor for post-AMI HF. After controlling for competitive risk events, the Fine-Gray (FG) model demonstrated that S100A8/A9 was associated with a higher incidence of post-AMI HF in the discovery cohort (HR: 2.03; 95% CI: 1.76–2.34; P < 0.001). In the subgroup analysis, S100A8/A9 levels were independently associated with HF across all pre-specified subgroups, including age, sex, various complications, and infarction severity (Supplementary Fig. 7), implying that S100A8/A9 had no interaction with these factors.

The predictive value of S100A8/A9 was compared to that of existing clinical biomarkers/variables. In the receiver operating characteristic analysis, S100A8/A9 demonstrated better predictive capabilities for HF than cTnI, BNP, and hs-CRP (Fig. 2c). Among the biomarkers, S100A8/A9 had the highest C-statistic (Supplementary Table 8). We further investigated whether adding S100A8/A9 to the known clinical variables would improve risk estimation. S100A8/A9 increased C-statistic significantly when added to the reference model (ΔC-statistic: 0.04 [95% CI: 0.02–0.05], P < 0.001) in the discovery cohort (Table 2). The addition of S100A8/A9 improved reclassification of the reference model (net reclassification index [NRI]: 0.23 [95% CI: 0.12–0.32], P < 0.001; HF NRI: 0.09 [95% CI: 0.01–0.12]; no-HF NRI: 0.14 [95% CI: 0.09–0.23]) (Table 2 and Supplementary Table 9). However, adding S100A12 to the reference model did not improve risk stratification. Collectively, S100A8/A9 provided incremental information on known clinical biomarkers/variables for HF prediction. We conducted decision curve analysis to gain a more precise understanding of the improvement in the model by adding S100A8/A9. Compared to the reference model, the model including S100A8/A9 would on average identify ~34 additional cases, without identifying any additional false positives, in a population of 1000 patients with an incidence of HF events of 28% (Supplementary Fig. 8a).

## Validation of the predictive value of S100A8/A9 for HF in validation cohort

Circulating S100A8/A9 levels were an independent HF predictor after adjusting for the same variables in the validation cohort (HR: 2.15 [95% CI: 1.79–2.58], P < 0.001), confirming S100A8/A9 as a robust HF predictor (Fig. 2d, e). After controlling for competitive risk events, the FG model showed that S100A8/A9 was associated with a higher incidence of post-AMI HF in the validation cohort (HR: 2.14; 95% CI:1.76–2.60; P < 0.001). Additionally, S100A8/A9 better predicted HF than cTnI, BNP, and hs-CRP (Fig. 2f and Supplementary Table 10). Furthermore, consistent with the results from the discovery cohort, adding S100A8/A9 to the reference model also improved risk stratification (Table 2 and Supplementary Table 11). Compared to the reference model, the model, including S100A8/A9, would on average identify ~19 additional cases, without identifying any additional false positives, in a population of 1000 patients with an 18% incidence of HF events (Supplementary Fig. 8b).

The Killip class is a specialized indicator of cardiac function in patients with AMI. We observed that the S100A8/A9 concentrations were higher in Killip class II and III patients than in Killip class I patients in the combined cohort (Supplementary Fig. 9). Due to the limited

**Table 2 | Discrimination/reclassification improvement by S100A8/A9 and S100A12 for HF risk prediction**

| | C-statistic (95% CI) | △C-statistic (95% CI) | P-value | NRI (95% CI) | P-value |
|---|---|---|---|---|---|
| **Discovery cohort** | | | | | |
| S100A8/A9 | 0.65 (0.62–0.68) | | | | |
| S100A12 | 0.53 (0.49–0.56) | | | | |
| Reference model | 0.70 (0.67–0.73) | Reference | | Reference | |
| +S100A8/A9 | 0.74 (0.71–0.77) | 0.04 (0.02–0.05) | 2.99E-06 | 0.23 (0.12–0.32)<br>Non-Events: 0.14 (0.09–0.23)<br>Events: 0.09 (0.01–0.12) | <0.001 |
| +S100A12 | 0.70 (0.67–0.73) | −0.001 (−0.003–0.002) | 0.698 | −0.01 (−0.03–0.04)<br>Non-Events: −0.001 (−0.016–0.026)<br>Events: −0.007 (−0.029–0.026) | 0.493 |
| **Validation cohort** | | | | | |
| S100A8/A9 | 0.70 (0.66–0.74) | | | | |
| Reference model | 0.79 (0.76–0.82) | Reference | | Reference | |
| +S100A8/A9 | 0.82 (0.78–0.85) | 0.03 (0.01–0.05) | 0.001 | 0.10 (0.03–0.23)<br>Non-Events: 0.052 (0.006–0.100)<br>Events: 0.052 (0.002–0.154) | 0.009 |

ΔC-statistic indicated the difference in comparison to the "reference model."
The reference model included age, sex, systolic blood pressure, Killip classification at admission, fasting glucose, creatinine, left main artery disease, neutrophil count, cTnI, BNP, hs-CRP, left ventricular ejection fraction at admission, and estimated infarct size (CK-MB $AUC_{0-72}$).
The added predictive ability of the candidate biomarkers beyond that of the reference model was assessed using Harrell's concordance C-statistic calculated from the Cox regression model and logistic model-based categorical NRI. NRI was assessed using the two category-based NRIs using 10% and 30% as HF-event cut-offs to define patient subgroups at low, intermediate, or high risk. All tests were two-sided and $P < 0.05$ was considered as significant. Source data are provided in the Source Data File.

sample size in patients with Killip class III (20 and 14 in discovery and validation cohort), a rising trend was only observed between Killip class III and class II.

The follow-up time for both the discovery and validation cohorts extends beyond 2020, presenting the possibility of confounding from the COVID-19 pandemic on HF risk[12]. We performed a sensitivity analysis of the association between S100A8/A9 and HF before and after COVID-19 pandemic (January 1, 2020). S100A8/A9 levels remained independently associated with HF risk (Supplementary Table 12).

Kaplan–Meier curves illustrated that patients in the higher risk categories stratified by the quartile of S100A8/A9 exhibited a higher risk of post-AMI HF events in both cohorts. The cutoff values of S100A8/A9 in the high-risk group were 5059 ng/mL and 4877 ng/mL in the discovery and validation cohorts, respectively, suggesting that a S100A8/A9 level exceeding 5000 ng/mL may be a reference for a higher risk of HF (Fig. 3a, b). Supplementary Table 13 illustrates S100A8/A9 plasma concentrations in relation to all the clinical characteristics.

**Causal relationship between S100A8/A9 and HF**
We next explored whether elevated S100A8/A9 levels drive HF development. The causal effects of HF-associated proteins were firstly investigated using individual-level one-sample MR analysis. To establish genetic instruments associated with the S100A8/A9 level, we search for genome-wide association studies (GWAS) summary statistics of plasma proteins for all 4907 aptamers at https://www.Decode.com/summarydata/. However, we could not find GWAS summary statistics for S100A8/A9 or S100A8. Because S100A9 regulates S100A8/A9 complex functions through various mechanisms, including protecting S100A8 from degradation, and given that S100A9 levels were strongly correlated with S100A8/A9 levels in the plasma ($r = 0.92$, $P < 0.01$, Supplementary Fig. 10), we selected 50 *cis*-protein quantitative single nucleotide polymorphisms (SNPs) of S100A9 from a GWAS of 35,559 Icelanders. The 24 SNPs with beta > 0 were associated with elevated S100A8/A9 concentrations, and 26 SNPs with beta < 0 were associated with decreased S100A8/A9 concentrations (Supplementary Table 14). Among the 50 protein quantitative trait loci (pQTLs), 20 with minor allele frequency >0.01 in East Asians were used for linkage disequilibrium analysis, and six

tag SNPs ($r^2 < 0.8$) were included in the S100A8/A9 genetic score (Supplementary Fig. 11). The genotype-tissue expression (GTEx) portal strongly supported that 17/20 pQTLs were associated with differential S100A8/A9 mRNA expression (Supplementary Tables 15 and 16). Because estimated effect of the SNPs on S100A8/A9 levels were obtained in Icelanders, and the genetic backgrounds of Icelanders and Chinese were heterogeneous, we confirmed the association between the genetic score and S100A8/A9 levels in a general Chinese population. Among these 588 HCs, the increase in the S100A8/A9 genetic score was significantly associated with the high-risk S100A8/A9 levels (refer to S100A8/A9 levels are greater than the mean plus one standard deviation of this cohort) (odds ratio [OR] per SD: 1.40 [95% CI: 1.11–1.76], $P = 0.004$). Additionally, the association was not affected by age and sex adjustment (OR per SD: 1.39 [95% CI: 1.11–1.75]; $P = 0.005$).

We then conducted genotyping for the six tag SNPs and calculated the S100A8/A9 genetic score for each individual in the validation cohort. Rs12033317 ($\beta = 0.33$) and rs12119788 ($\beta = 0.89$) were associated with increased S100A8/A9 levels, and rs1560832 ($\beta = -0.52$), rs3014874 ($\beta = -0.47$), rs3014875($\beta = -0.42$), and rs59961408 ($\beta = -0.16$) were associated with decreased S100A8/A9 levels, consistent with the directions in the GWAS summary statistics. A scatter plot illustrated the positive association between the genetic score and S100A8/A9 plasma concentrations for validation cohort (Supplementary Fig. 12). For a 1-SD increase in S100A8/A9 genetic scorers, the incidence of S100A8/A9 levels exceeding 4877 ng/mL (cutoff value for high risk of HF) increased by 43%, both in raw data and after adjusting for sex and age. We also provided a Kaplan–Meier curve comparing the S100A8/A9 genetic score to HF (Fig. 3c). At the median follow-up time, the patients in the higher risk categories stratified by the quartile of S100A8/A9 genetic score exhibited a higher risk of post-AMI HF events. The stratification was less significant due to the limited patients at the late follow-up time. We subsequently performed one-sample MR and observed that genetically predicted S100A8/A9 values were associated with HF (OR per SD: 1.20 [95% CI: 1.003–1.44], $P = 0.047$) after adjusting for age, sex, systolic blood pressure, Killip classification at admission, fasting glucose, creatinine, left main artery disease, neutrophil count, cTnI, BNP, CRP, LVEF at admission, and estimated infarct size (CK-MB $AUC_{0-72}$).

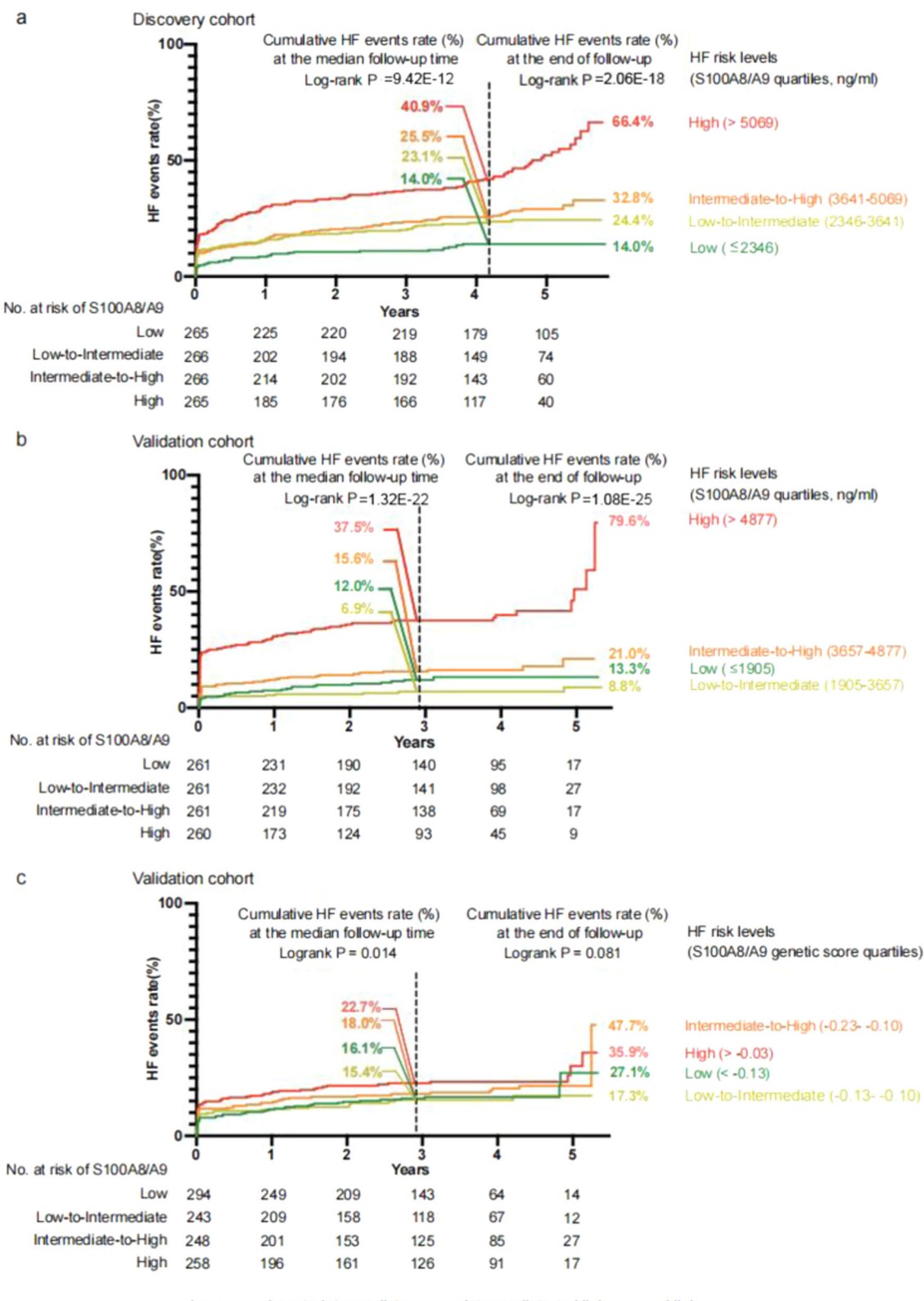

**Fig. 3 | Incidence of HF events according to S100A8/A9 levels and S100A8/A9 genetic score.** Kaplan–Meier curves illustrate the timing of HF events in the four strata of S100A8/A9 levels and S100A8/A9 genetic score. The quartile of S100A8/A9 (ng/ml) was used to classify patients into four risk categories, including low, low-intermediate, intermediate-high, and high-risk in the discovery cohort (**a**) and validation cohort (**b**). Similarly, the quartile of the S100A8/A9 genetic score was used to classify patients in the validation cohort into four risk categories (**c**). The vertical black dashed line means the median follow-up time (4.2 and 2.9 years for discovery and validation cohorts respectively). The maximum follow-up time is 5.8 years and 5.3 years for discovery and validation cohorts respectively. *P*-values reported are two-tailed from log-rank tests. *P* < 0.05 was considered as significant. Source data are provided in the Source Data File.

**Table 3 | Association of genetically determined plasma S100A8/A9 with risk of HF events in one- and two-sample MR analysis**

| Methods | Genetic tools | OR (95%CI) | *P*-value |
|---|---|---|---|
| One-sample MR | | | |
| 2SLS | S100A8/A9 genetic score | 1.20 (1.003–1.44) | 0.047 |
| Two-sample MR | | | |
| Post-AMI HF | | | |
| Wald ratio | S100A8/A9 genetic score | 2.06 (1.25–3.39) | 0.004 |
| IVW | cis-pQTLs for S100A8/A9 (rs59961408, rs1560832, rs3014874, rs3014875, rs12033317) | 1.55 (1.15–2.09) | 0.004 |
| MR-Egger regression (intercept = −0.10, *P*-value = 0.744) | | | |
| General HF | | | |
| IVW | cis-pQTLs for S100A8/A9 (rs59961408, rs12119788, rs1560832, rs3014874, rs3014875, rs12033317) | 1.04 (1.01–1.07) | 0.016 |
| MR-Egger regression (intercept = 0.001, *P*-value = 0.894) | | | |

The two-stage least squares method was used to calculate causal effects in the one-sample MR. The causal effect was calculated by combining variant-specific causal estimates in an IVW fixed-effects meta-analysis in two-sample MR. All tests were two-sided and *P* < 0.05 was considered as significant. Source data are provided in the Source Data File.
*CI* confidence interval, *HF* heart failure, *IV* instrumental variable, *IVW* inverse-variance weighted, *MR* Mendelian randomization, *OR* odds ratio, *SNPs* single nucleotide polymorphisms, *UKB* UK Biobank, *2SLS* Two Stages Least Square.

To confirm this result, we conducted a two-sample MR analysis and verified the causal effect of S100A8/A9 on post-AMI HF (Supplementary Fig. 13). In 1114 patients with AMI from the UK Biobank (UKB) database (Supplementary Table 17), owing to the absence of rs12119788, S100A8/A9 genetic scores comprised the other five SNPs. For a 1-SD increase in S100A8/A9 genetic scorers, the risk of HF increased by 30% (raw) and 29% (adjusted for sex and age). Two-sample MR analysis showed a genetically instrumented per-SD S100A8/A9 was associated with higher odds of post-AMI HF in Wald ratio analysis (OR: 2.06 [95% CI: 1.25–3.39]; *P* = 0.004) and inverse-variance-weighted (IVW) analysis (OR: 1.55 [95% CI: 1.15–2.09]; *P* = 0.004) (Table 3). The Egger method provided no evidence of pleiotropy (Egger's intercept, *P* = 0.744).

We further validated this finding in general HF using data from the finn-b-I9_HEARTFAIL study (Supplementary Fig. 13). Using cis-pQTLs effect estimates on S100A8/A9 levels from the validation cohort, increased genetically predicted S100A8/A9 levels were associated with increased risk of HF (IVW estimate of OR per SD: 1.04 [95% CI: 1.01–1.07]; *P* = 0.016) (Table 3). We provided dose-response curves showing the causal effect of S100A8/A9 levels on post-AMI/general HF, with a line of best drawn (Supplementary Fig. 14). The slope is the causal estimate of S100A8/A9 levels on post-MI/general HF.

Colocalization analyses can strengthen the evidence for a causal effect. However, guidelines for colocalization analysis recommend only testing for colocalization where *P* < 10⁻⁶ for both traits in question. Due to the small sample size of the post-AMI cohort, there is insufficient power to test colocalization in our cohort. Although the sample size for general HF is much larger, the minimum *P*-value of the SNPs within the target region is 0.0068, which does not meet the criteria for conducting colocalization analysis. However, we provided a visual comparison of the pQTL and GWAS signals at the locus for both UKB cohort and finn-b-19_HEARTFAIL cohort by showing LocusZoom plots of the pQTL and GWAS signals side-by-side (Supplementary Fig. 15). The plots showed that the pQTL and GWAS for post-AMI HF signals were all around the S100A9 gene.

## Discussion

In our proteomic analyses, high inflammation status, particularly increased S100A8/A9 level, was observed in patients with post-AMI HF. In two independent prospective cohorts, S100A8/A9 robustly predicted HF. We demonstrated that a genetically determined portion of S100A8/A9 was associated with post-AMI HF risk, suggesting that S100A8/A9 acts as an intermediate causal phenotype in post-AMI HF.

Although data regarding the association between high inflammatory status and post-AMI HF exist[13], the existing known biomarkers are insufficient for precise HF prediction in patients with AMI. High S100A8/A9 levels during the acute event were reportedly associated with increased hospitalization for HF during follow-up in patients with MI[14]. However, this study had certain limitations, including a small sample size (HF: *n* = 41, no-HF: *n* = 483) of patients with ACS, lack of validation, and no significant result after adjusting for multiple factors. Therefore, the prognostic value of S100A8/A9 in AMI needs to be further established. Our study revealed the prognostic value of S100A8/A9 in post-AMI HF in two independent prospective cohorts. Adjustments for age, sex, cardiovascular risk factors, and well-established risk indicators did not significantly affect the association between S100A8/A9 and post-AMI HF. Moreover, S100A8/A9 was superior to established biomarkers, including cTnI, BNP, and CRP, and added discrimination/reclassification value to a reference model of traditional and established risk indicators.

MR analyses of S100A8/A9 enhanced the significance of our compared to other observational studies on prognostic markers. MR can reveal novel the pathological mechanisms and predict novel drug targets for several cardiovascular diseases[15,16]. More recently, Lumbers et al. reported a large-scale MR analysis of incident HF that combined observational associations of 90 cardiovascular proteins with a systematic appraisal of causal effects. However, S100A8/A9 was not included in the Olink-Proseek Multiplex proximity extension assay for cardiovascular protein assessment, preventing the discovery of a correlation or causal relationship between S100A8/A9 and HF[17]. Therefore, our study is the first to demonstrate the causal effect of S100A8/A9 on post-AMI HF. Because of the close correlation between S100A8 and S100A9 levels and the dependence of S100A8 protein stability on S100A9 expression[18], we chose *cis*-variants, as opposed to *trans*-variants, of S100A9 as genetic instruments. In this regard, the scope for violating the exclusion restriction assumption was limited because the pQTL variant effects on the outcome were possibly mediated through the expression of the protein under consideration (no horizontal pleiotropy)[19]. The association between genetic variants and exposure was strongly supported by the GTEx portal, suggesting the effectiveness of S100A9 genetic instruments. Among the 20 SNPs shown in Supplementary Table 14, rs3014874 was both a pQTL and an expression quantitative trait locus with the most significant *P*-value. Rs3014874 may be a functional variant because of its location in the distance enhancer E1385341. Additionally, rs12119788 and rs2070864 were located in a promoter-like element, E1385328, and another distance enhancer, E1385333, implying that the target region has multiple functional SNPs.

In older patients, the prognosis of AMI is influenced by many other diseases, and the role of genetics in S100A8/A9 levels is also influenced by more confounding factors. Therefore, we selected patients with early-onset AMI in the validation cohort to reduce confounding influences and highlight the role of genetic variants. The validation cohort comprised unusually young patients with AMI. To exclude the possibility that the observed causal effect of S100A8/A9 on HF risk only applies to patients with early-onset MI, we used the UKB cohort for external validation of the causal effect, in which the median age at AMI onset was 56.0 [52.6–59.2] in patients with HF and 55.7 [51.6–58.4] in patients without HF (Supplementary Table 17). Additionally, there is a causal effect of S100A8/A9 on post-AMI HF in the UKB cohort, indicating that the observed causal effect of S100A8/A9 on HF risk is not limited to individuals with early-onset MI.

The pathological role of S100A8/A9 has been established in experimental ischemia/ischemia-reperfusion injury[14,20,21]. Mitochondrial dysfunction and oxidative stress can cause cardiomyocyte death. S100A8/A9 directly induces mitochondrial dysfunction by suppressing mitochondrial complex I activity[20]. S100A8/A9 interacts with NADPH oxidase complex by binding to p67phox and Rac in neutrophils, thereby promoting oxidative stress[22]. Persistent and excessive inflammation might contribute to myocardial injury aggravation and adverse cardiac remodeling[23]. During ischemia/ischemia-reperfusion injury, S100A8/A9 promoted cardiac inflammatory response by stimulating leukocyte activation/infiltration, amplifying NF-κB signaling activation, and proinflammatory cytokine secretion[14,20,21,24]. Additionally, microvascular obstruction caused by thrombosis contributed to infarct extension after percutaneous coronary intervention[25]. S100A8/A9 modulated platelet function and promoted thrombus formation[26]. Collectively, S100A8/A9 is an important driver of various pathological processes during ischemia/ischemia-reperfusion injury. These functions demonstrate that S100A8/A9 can be used as a strong predictor of post-AMI HF and support its causal effect in the MR analysis.

Although neutrophils express S100A8/A9/12, there are several possible explanations for the improved predictive ability of S100A8/A9 compared to S100A12. First, the S100A8/A9 levels in both the serum and PBC were much higher than the S100A12 levels in patients with AMI. Second, once secreted, extracellular S100A8/A9 and S100A12 function by activating the pattern recognition receptors, including TLR-4 or receptor for advanced glycation end-products (RAGE)[27]. We previously showed that S100A8/A9-induced cardiomyocyte death depended on TLR-4, rather than RAGE[20]. Moreover, S100A8/A9 promoted granulopoiesis by interacting with TLR-4, subsequently priming the inflammasome[21]. RAGE is the target protein of S100A12[28], and in vivo affinity of S100A12 to RAGE is higher than that of the S100 protein family[29]. Moreover, there are insufficient studies on the direct effect of S100A12 on ischemia/ischemia-reperfusion injury, though S100A12 promotes atherosclerosis and vascular calcification[30].

Measuring S100A8/A9 levels at admission may identify patients at high risk of post-AMI and is suitable for the early use of medications and careful post-discharge follow-up. Elevated S100A8/A9 levels drive the post-AMI progression of the inflammatory response and mitochondrial dysfunction. Several anti-inflammatory agents or mitochondria-targeting peptides are promising drugs for cardiovascular disease[31,32], and these treatments may be more effective for patients with high S100A8/A9 levels. S100A8/A9 levels are elevated before overt HF, and S100A9 blockade reportedly improves cardiac function in experimental MI and ischemia-reperfusion[14,20,21]. In present study, we prove the causal effect of S100A8/A9 on post-AMI HF in human. Both human and experimental study indicate that S100A8/A9 is a promising therapeutic target for post-AMI HF. Administration with ABR238901(an orally active and potent S100A8/A9 blocker) or a S100a9 neutralizing antibody, could prevent cardiac injury and HF post-AMI in animal model[14,20]. Collectively, future randomized controlled trials are required to elucidate whether targeting S100A8/A9 have clinical benefits in post-AMI HF prevention.

This study had some limitations. First, it was completed at a single center. therefore, further research is required using multicenter prospective cohorts. However, the consistency in the prognostic value of S100A8/A9 in the two independent cohorts suggested that S100A8/A9 is a reliable predictive biomarker. Second, S100A8/A9 was measured at admission, and a repeated-measures analysis may have captured a larger proportion of the S100A8/A9 variance. However, this early time point is the most useful for identifying a possible biomarker-based risk stratification. Third, due to the high cost of protein profiling experiments, we performed the initial screening step in a small number of subjects. As a result, other causal proteins may have been missed due to the low detection power in the initial screening step. Lastly, the urine albumin-creatinine ratio (UACR) is a prognostic marker of adverse HF outcomes in patients with ACS with type 2 diabetes[33,34]. We have not conducted this analysis because of the missing of UACR in present study. However, in our discovery and validation cohorts, 67% and 80% patients have no history of type 2 diabetes, respectively. Collectively, our major findings remain solid and robust despite these limitations.

In conclusion, high S100A8/A9 levels are associated with an increased post-AMI HF risk, providing an efficient approach for identifying patients at high HF risk. MR analysis revealed a causal effect of S100A8/A9 on post-AMI HF, supporting the possible role of anti-S100A8/A9 interventions in HF prevention. S100A8/A9 can improve post-AMI HF risk stratification, and its causal effects can help elucidate the pathological mechanisms of post-AMI HF.

## Methods

The written informed consent was obtained from all the participants, and the Ethics Committee of Beijing Anzhen Hospital Capital Medical University approved the study protocol (approval number: 2018010). All patients participated in the study process without compensation. This study followed the Strengthening the Reporting of Observational Studies in Epidemiology reporting guideline[35] and the Declaration of Helsinki and was registered at ClinicalTrials.gov (ID: NCT03752515).

### Study design

This study comprised the following three steps (Fig. 1): (1) HF-related candidate proteins were identified using serum proteomics in a cross-sectional set of patients with AMI who developed HF during hospitalization, patients with AMI without HF, and HCs. (2) The association between candidate proteins and HF was prospectively evaluated in the discovery (HF: $n = 296$, no-HF: $n = 766$) and validation (HF: $n = 192$, no-HF: $n = 851$) cohorts. (3) The causal relationship between HF-associated proteins and HF was confirmed using MR analysis. Genetic instruments associated with the S100A8/A9 level were identified from published GWAS[36]. For individual-level one-sample MR analysis, the causal association of genetic instruments with post-AMI HF events was assessed in the validation cohort (HF: $n = 192$, no-HF: $n = 851$). For the two-sample MR analysis, estimates of the association between the genetic instruments and S100A8/A9 levels from the validation cohort and the association between the genetic instruments and post-AMI HF from the UKB cohort were used to examine the causal effect of S100A8/A9 levels on post-AMI HF. Moreover, a statistical summary of the association between genetic instruments and S100A9 levels from the validation cohort and a statistical summary of the GWAS from the finn-b-I9_HEARTFAIL study were used to evaluate the causal association between genetic instruments and general HF.

### Study participants and sampling

For the discovery cohort, we recruited consecutive 1324 patients with AMI at the Beijing Anzhen Hospital of Capital Medical University

between August 1, 2015, and November 30, 2017. Patients with cardiogenic shock (Killip class IV), active infection, systemic inflammatory disease, known malignant disease, or surgery within the previous 3 months were excluded, while 1,062 patients were enrolled according to stringent criteria (Supplementary Fig. 1).

For the validation cohort, we recruited 1183 patients with AMI (age: 18–45 years) from Beijing Anzhen Hospital of Capital Medical University between February 1, 2016, and January 30, 2020, with similar exclusion criteria, and finally included 1043 patients with AMI (Supplementary Fig. 1).

HCs were recruited among individuals receiving regular physical examinations at Beijing Anzhen Hospital, and HC status was confirmed using electrocardiogram, transthoracic echocardiography, and laboratory examination to exclude MI, HF, or abnormal cardiac structure/function.

For patients with AMI from the UKB cohort, we used data collected at the UKB assessment centers at baseline, combined with information on incident disease events from the hospital and death registry. Notably, 1144 individuals diagnosed with MI and <60 years of age were included in our study. The UKB study was approved by the North West Multi-Center Research Ethics Committee, and all participants provided written informed consent to participate in the UKB study. Post-AMI HF includes HF and mortality due to cardiovascular disease (CVD). HF was defined using the ICD-10 code I50. Death due to CVD was defined using the ICD-10 codes for different endpoints from the Death Registry.

## Study definitions
AMI was defined as continuous chest pain for >30 min, new ischemic electrocardiogram changes, and elevated cTnI levels with at least one value above the 99th percentile upper reference limit[37]. Two independent cardiologists diagnosed each patient. In-hospital HF included new HF onset (HF symptoms/signs after initial presentation and imaging evidence of pulmonary congestion), worsening HF (Killip class II progressing to III or IV, and Killip class III progressing to IV), cardiogenic shock diagnosis, and in-hospital death due to HF or cardiogenic shock. Long-term HF included HF progression resulting in rehospitalization and/or death due to HF after the initial discharge. Supplementary Table 3 presents the specific definitions of each event type. All HF events were adjudicated by the consensus of two experienced cardiologists who were blinded to the study results by review of outpatient clinics or hospitalization records and telephone interview. Any disagreements were resolved through discussion and by seeking a third opinion from another blinded, experienced cardiologist as required. Current smokers were defined as those who have smoked 100 cigarettes in their lifetime and had smoked cigarettes in the past 30 days.

## Study outcomes and follow-up
The study outcomes were the in-hospital and long-term post-discharge HF incidence. Patients were followed up at 6–12-month intervals for HF events by telephone interviews and review of outpatient clinics or hospitalization records. During follow-up, 118 in-hospital (11%) and 178 long-term (17%) HF events were recorded in the discovery cohort, and 110 in-hospital (11%) and 82 long-term (8%) HF events were recorded in the validation cohort. The follow-up ended at death or termination (May 31, 2021; the maximum follow-up for both the discovery and validation cohorts). The median follow-up times were 4.2 years (interquartile range [IQR]: 1.7–5.1) and 2.9 years (IQR: 1.6–4.2) in the discovery and validation cohorts, and 52 and 44 patients were lost to follow-up, respectively.

## Blood sample collection
Venous blood samples were collected on admission. The samples were placed into gel-containing vacutainer tubes and centrifuged within 1 h at 1800 g for 25 min; the serum, plasma, and blood cells were stored at −80 °C until use.

## Serum proteomics and analysis
**Human antibody array.** A human antibody array (a combination of Human L-507 and Human L-493, RayBiotech Inc.) was performed on the serum of HCs and patients with AMI with or without HF events at admission ($n = 10$ per group), according to the manufacturer's instructions. Each serum sample was hybridized to the arrays overnight at 4 °C. All slides were scanned using a GenePix 4000 B Microarray Scanner and analyzed using GenePix Pro 6.0 software. The protein levels were normalized to the internal controls.

**Selection of prognostic candidates.** DEPs were identified based on a P-value < 0.05 in AMI vs. HC or no-HF events vs. HF events. DEPs at the intersection of the two comparisons were identified as HF-related DEPs. Based on the HF-related DEPs, Reactome enrichment analysis was used to determine the pathways that were significantly associated with HF. The least absolute shrinkage and selection operator (LASSO) is a regression analysis method that minimizes the sum of least squares in a linear regression model and shrinks the selected beta coefficients using penalties[38], by incorporating shrinkage, this method provides a rigid variable selection and coefficient estimation. LASSO analysis excludes the least informative variables and selects the features of greatest importance for the outcome of interest in the imputed dataset. Among the selected HF-related proteins, LASSO analysis was conducted with the glmnet package in R, with a 10-fold cross-validation step to define the λ parameter that resulted in the minimum value of the mean square error of the regression model[39].

## Biochemistry measurement
Serum S100A8/A9 levels were measured using a standard enzyme-linked immunosorbent assay (ELISA) kit (S100A8/9; R&D Systems Europe, Abingdon, Oxford, UK). The assay employed a quantitative sandwich enzyme immunoassay technique. Briefly, a monoclonal antibody specific to the human S100A8/S100A9 heterodimer was precoated onto a microplate. Subsequently, the samples were pipetted into the wells. After washing away the unbound substances, an enzyme-linked monoclonal antibody specific for the human S100A8/S100A9 heterodimer was added to each well. All steps strictly followed the operation procedure of ELISA protocol. S100A12 levels were measured using a standard ELISA kit (S100A12: RayBiotech, Norcross, GA, USA). S100A9 levels were measured using a standard ELISA kit (S100A9: RayBiotech, Norcross, GA, USA).

All samples were assessed in duplicate in a blinded manner. The inter- and intra-assay coefficients of variation in patients with AMI were 3.62% and 3.37% respectively for S100A8/A9, 4.73% and 4.51% for S100A12, and 3.63% and 3.91% for S100A9. For randomly selected samples, the levels of biomarkers in fresh samples at admission correlated well with those in the same samples stored for 6 months (S100A8/A9: $r = 0.974$, $P < 0.001$; S100A12: $r = 0.961$, $P < 0.001$, S100A9: $r = 0.975$, $P < 0.001$), suggesting that storage at −80 °C for up to 6 months did not significantly affect the stability of serum S100A8/A9, S100A12 and S100A9.

cTnI, BNP, and hs-CRP levels were measured simultaneously with S100A8/A9 levels in both cohorts. cTnI, BNP, and hs-CRP levels were analyzed using the same assays in the two cohorts. Serum cTnI levels were determined using a chemiluminescence assay (Beckman Coulter, Access AccuTnI+3), and the 99th percentile of cTnI was 0.04 ng/mL. Plasma BNP levels were assessed using an Alere Triage immunoassay and read on an automated DxI800 platform (Beckman Coulter Diagnostics); normal levels were considered to be <100 pg/mL. Plasma hs-CRP levels were measured using a turbid metric inhibition immunoassay (Beckman Coulter AU5800 automatic biochemical analyser); normal levels were considered <3 mg/L. Serum cholesterol, glucose,

and creatinine levels were measured using routine laboratory methods.

## Infarct size estimation

The infarct size was evaluated using the area under the curve (AUC) for the CK-MB enzyme over the first 72 h after admission (CK-MB $AUC_{0-72}$)[40]. CK-MB levels were recorded at baseline (time of enrolment) and at 6, 24, 48, and 72 h after enrolment. The CK-MB measurements were performed in a central laboratory. The AUC was calculated using the linear trapezoidal method[41] based on the five CK-MB values (Supplementary Fig. 16).

## Genotyping and quality control in the validation cohort

Human genomic DNA was isolated from EDTA-anticoagulated blood using the proteinase K methods[42]. SNPs were genotyped in all populations using TaqMan technology according to the manufacturer's protocol. Negative controls without DNA were used for each plate to ensure no contamination between the samples and genotyping reagents. Genotypes that could not be automatically measured using Sequence Detection System 2.1 software were excluded. Direct Sanger sequencing was used to confirm the accuracy of genotyping.

## Genetic instruments

We downloaded published GWAS from >30,000 individuals[36] to identify the pQTLs of S100A8/A9. Because no record of S100A8 or S100A8/A9 was found in these public data, and given that S100A8 protein stability depends on S100A9 expression, we selected cis-pQTLs within 20 kb on either side of the *S100A9* gene (153357854-153361023 [grch38.p14]). Subsequently, SNPs with minor allele frequency >0.01 in East Asians (1000 Genome Project) were used for linkage disequilibrium analysis using Haploview software. The S100A8/A9 genetic score was constructed by combining variants in low-linkage disequilibrium ($r^2 < 0.8$) with all other variants. Supplementary Table 14 shows that rs12119788 and rs12033317 are associated with elevated S100A8/A9 plasma concentrations, while rs59961408, rs1560832, rs3014874, and rs3014875 are associated with decreased S100A8/A9 plasma concentrations.

## Sample size

To estimate the sample size required to assess the predictive value of candidate proteins, we assumed an event rate of 10% and covariate prediction of approximately 30% of the biomarker variance; a sample size of 913 patients provided 90% power (at $P < 0.05$) to detect a biomarker HR of 1.5. The number of patients included in this study met the inclusion criteria.

For MR, the sample size calculation was based on the results of an online tool using several parameters[43], including type-I error rate, power, OR of exposure and outcome, event rate, and variance explained by the selected genetic instruments.

## Statistical analyses

Data for categorical and continuous variables are presented as absolute numbers with percentages and medians with interquartile ranges (IQR: 25th–75th percentiles). $\chi^2$ or Fisher's exact test was used for categorical variables. Continuous variables were compared between the two groups using either Student's *t*-test or the Mann–Whitney *U*-test, as appropriate. Participants with missing data were excluded as they represented a small group. *P*-values < 0.05 were considered statistically significant. Analyses were performed using the IBM SPSS software (version 24.0; SPSS Inc., Chicago, IL, USA) or R (version 4.1.1; R Foundation for Statistical Computing, Vienna, Austria).

**Associations of candidate-protein levels with HF.** HF-related proteins were identified in the discovery cohort by univariate and multivariate analyses using a Cox regression model. Univariate models were used to identify potential confounders (Supplementary Table 7). Known risk factors (including sex) and potential confounders ($P < 0.05$ in univariate analysis) were included in the multivariate analysis. We constructed three multivariate models by adding variables for multivariate adjustment: model 1 (sex and age), model 2 (model 1 + systolic blood pressure [SBP], Killip classification at admission, fasting glucose, creatinine, and left main artery disease), and model 3 (model 2 + neutrophil count, cTnI, BNP, CRP, LVEF, and estimated infarct size [CK-MB $AUC_{0-72}$]).

The proportional-hazard assumption was assessed for time-to-event outcomes using the Schoenfeld residuals test, and no proportional-hazard assumption was violated for the biomarker variables. For continuous variables, HRs and the corresponding 95% confidence intervals (CIs) per 1-SD higher measure were calculated. Spline regression models were used to explore the shapes of the associations between the prognostic biomarkers and outcomes by fitting a restricted cubic spline function[44]. The analyses were multivariate-adjusted and used three knots (5th, 50th, and 95th percentiles). We examined the association between candidate proteins and time-to-event ratios in the different subgroups. This approach allowed us to estimate subgroup-specific HR and compare the HRs in the two categories of differing subgroup variables. Kaplan–Meier cumulative-event curves were used to display outcomes independent of prognostic biomarker-guided risk status; group-wise comparisons were based on the log-rank test.

As the rate of an event and the corresponding event risk do not have a direct relationship under the presence of competing events, the Fine-Gray competing risk model based on the R package "cmprsk" was used to examine the associations while accounting for death (insufficient evidence to determine death from HF, non-cardiac death) as a competing risk. The Fine-Gray model computes the HR adjusted for sex, age, SBP, Killip classification at admission, fasting glucose, creatinine, left main artery disease, neutrophil count, cTnI, BNP, CRP, LVEF at admission, and estimated infarct size (CK-MB $AUC_{0-72}$)[45].

**Incremental predictive value of candidate proteins.** Since no established/reliable post-MI HF prediction models exist, we constructed reference models using risk factors with $P < 0.05$ in the univariate analysis (age, sex, SBP, Killip classification at admission, fasting glucose, creatinine, left main artery disease, neutrophil count, cTnI, BNP, CRP, LVEF at admission, and estimated infarct size (CK-MB $AUC_{0-72}$)). The added predictive ability of the candidate biomarkers beyond that of the reference model was assessed using Harrell's concordance C-statistic calculated from the Cox regression model and logistic model-based categorical NRI. Harrell's concordance C-statistic and time-dependent receiver operating characteristic (ROC) curve analysis were used to compare the predictive accuracy of the candidate proteins and clinical biomarkers. The clinical usefulness of S100A8/A9 was evaluated using decision curve analysis (DCA)[46], by estimating the net benefit of adding S100A8/A9 to risk stratify patients according to different decision thresholds of HF-event risk compared with the reference model.

**Causality between S100A8/A9 and HF.** Individual-level one-sample and two-sample MR analyses were used to investigate the causality between S100A8/A9 and HF incidence. Three general assumptions of MR were applied: (i) robust S100A8/A9 association, (ii) no association with HF confounders, and (iii) association only with HF based on their effect on S100A8/A9.

## One-sample MR analysis

Individual-level one-sample MR analysis was used to investigate causality between S100A8/A9 and HF. For one-sample MR analysis, the

S100A8/A9 genetic score was calculated for each participant in the validation cohort by summing the number of effect alleles that a participant inherited at each variant in the score, weighted by the effect of each variant on S100A9 levels (β value in the GWAS summary). We used a two-stage least squares method to regress S100A8/A9 levels on the S100A8/A9 genetic score and used estimation prediction to generate a genetically predicted plasma S100A8/A9 value for each validation cohort participant. These values were tested for HF risk associations using multivariable regression. The analyses were performed using the R package "ivreg".

### Two-sample MR analysis to assess the causal effect of S100A8/A9 on post-AMI HF

Two-sample MR analysis is less prone to false-positive bias, which is possible in one-sample MR analysis[47]. We obtained the effect estimates of the S100A8/A9 genetic score on S100A8/A9 levels in the validation cohort and the effect estimates of the S100A8/A9 genetic score on post-AMI HF in the UKB (1144 patients with AMI). We estimated the causal effect of S100A8/A9 levels on post-AMI HF using the Wald ratio. Furthermore, we calculated causal effects by combining variant-specific causal estimates in an IVW fixed-effects meta-analysis. Specifically, we obtained the effect estimates of the five SNPs on S100A8/A9 levels in the validation cohort and the effect estimates of the five SNPs on post-AMI HF from the UKB (1144 patients with AMI). Subsequently, we estimated the causal effect of S100A8/A9 levels on post-AMI HF using the Wald ratio and used the IVW method to generate an estimate of all SNPs. Pleiotropy was assessed using the MR-Egger regression.

### Two-sample MR analysis to assess the causal effect of S100A8/A9 on general HF

To determine the effect size of SNPs on S100A8/A9, we used association statistics between the six SNPs and S100A9 levels in a validation cohort of 1043 patients with AMI. We used an association statistical summary between the six SNPs and HF from the finn-b-I9_HEARTFAIL study (13,087 patients with HF and 19,5091 controls) to obtain the effect size estimates of the association between target SNPs and HF. Causal-effect estimates of S100A8/A9 levels on HF events were obtained separately for each SNP by Wald ratio[48], and to generate an estimate using all the SNPs, we used the inverse-variance-weighted method for primary analysis and MR-Egger for sensitivity analysis.

### Reporting summary

Further information on research design is available in the Nature Portfolio Reporting Summary linked to this article.

## Data availability

The raw data supporting the findings of the study have been provided in source data. Due to ethical and legal restrictions, the clinical data are not publicly available. Any individual affiliated with an academic institution may request access to the clinical data from Yulin Li, PhD (lyllyl_1111@163.com) for research purposes. This includes submitting a proposal to the management team, where upon approval, data will be provided with a signed data access agreement. The timeframe for responding to an access to information is a 20-working day from the date of receipt. Source data are provided with this paper. Three public data were used in the MR analysis. The GWAS summary statistics for S100A9 "5339_49_S100A9_calgranulin_B.txt.gz" be downloaded at https://www.decode.com/summarydata/. The data of AMI patients in UK Biobank cohort were download from at https://www.ukbiobank.ac.uk/ and received under the data request application no.68808. The HF GWAS of summary statistics from finn-b-I9_HEARTFAIL study are publicly available at https://gwas.mrcieu.ac.uk/datasets/finn-b-I9_HEARTFAIL/. Source data are provided with this paper.

## Code availability

We used publicly available software for the analyses, and all software used is listed and described in the Methods section of our manuscript. Statistical analyses were conducted in R statistical software. Proteome analysis was used the R glmnet package (https://cran.r-project.org/web/packages/glmnet/index.html) and pathway enrichment analyses were conducted using the clusterProfiler package in R (https://pubmed.ncbi.nlm.nih.gov/22455463/) and the ReactomePA R package (https://bioconductor.org/packages/release/bioc/html/ReactomePA.html). MR analyses were conducted using the TwoSampleMR package in R. (https://mrcieu.github.io/TwoSampleMR/), genetic colocalization analyses were conducted using the coloc package in R.

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

## Acknowledgements

We appreciate all the patients who participated and thank Dr. Jian Cui (Shanghai BioGenius Biotechnology Co., Ltd.) for providing bioinformatics assistance. We thank Dr. Jie Zhao (School of Public Health, University of Hong Kong) and Dr. Lan Liu (School of Statistics, University of Minnesota, Twin Cities) for their assistance on statistics. This study was funded by the National Science Foundation of China (82230013, 81770245, 81970215 to YL.L), Key Laboratory of Remodeling-Related Cardiovascular Diseases, Ministry of Education, China, Beijing Municipal Public Welfare Development and Reform Pilot Project for Medical Research Institutes (JYY2023-9), Beijing Municipal Health Commission (11000023T000002039525), and Beijing Hospitals Authority's Ascent Plan.

## Author contributions

L.Y.L. designed the study. J.M., P.L., Y.X.Y., F.G. and H.G. collected samples and completed the follow-up of AMI patients. J.M., Y.L., L.S.Z., K.M. and H.Z. performed the experiments and data analysis. L.Y.L. and Y.L. wrote the manuscript. L.X.M., J.D. and L.Y.L. organized and supervised the study.

## Competing interests

The authors declare no competing interests.
