## [Peer Review File · Nature Communications]

REVIEWER COMMENTS

Reviewer #1 (Remarks to the Author):

Ma et al. used protein array-based serum proteomics and single cell transcriptomics to select candidate biomarkers for post-acute myocardial Infarction Heart Failure (Post-AMI HF), then used Mendelian Randomization study to confirm the causal effects of S100A8/A9 in Post-AMI HF. The results may be interesting for predicting Post-AMI HF, however, there are several issues in the manuscript, as mentioned below, that should be clarified.

1) In this manuscript, the authors used protein array-based technology to detect the levels of 1000 proteins in three different groups. The authors detected both S100A8 and S100 A8/A9 as differential proteins across the three groups (Supplemental table 3). Though S100A8 preferentially exists as a heterodimer or heterotetramer with S100A9 as calprotectin (S100A8/A9), how about the level of S100A9 in the three groups? Could the level of S100A9 remain the same in the three groups, while the level of S100A8 increased significantly in HF group? Mass spectrometry-based proteomic technology, such as MRM-based targeted proteomic method, may provide more information about the level of S100A9 in the three groups.

2) In both discovery cohort and validation cohort, the authors detected the level of S100A8/A9 using ESILA and found that the level of S100A8/A9 was increased in HF patients compared to no-HF patients (Supplemental Figure 8 and Figure 13). I am wondering whether higher level of S100A8/A9 definitely indicates a higher risk of Post-AMI HF? or whether there is a concentration range of S100A8/A9 which could be as a reference for a higher risk of Post-AMI HF?

3) The authors states that “If a biomarker is causally associated with HF, it not only helps the prediction of HF, but also serves as the target of intervention”(line 69-71), could S100A8/A9 be used as target for intervention of Post-AMI HF and how S100A8/A9 could be used as intervention of post-AMI HF?

Reviewer #2 (Remarks to the Author):

I am reviewing for the first time the article by Ma and colleagues which aims to identify independent markers of inflammation in a post-MI population. The authors identified S100A8/A9 as a potential causative marker associated with HF in post-MI populations.

This is an overall well designed study with unique insights.

I have the following comments:

1. Did heart failure outcomes include future assessment of LVEF? This appears to be a baseline covariate but not recorded in the future
2. The incremental improvement in c-statistic is limited in the validation cohorts – i.e. from 0.7 to 0.73 from reference model to model that includes S100A8/A9. Can the authors corroborate whether they feel this is a significant improvement? If so, then why?

3. Use of mineralocorticoid receptor antagonist (MRA) such as spironolactone are evidence based in a post MI population. Could needs to be included in baseline tables and also in the baseline model as a sensitivity analysis

4. In post-MI populations, urine albumin-creatinine ratio is a very strong prognostic marker of adverse HF outcomes. See Razaghizad, A., Sharma, A., Ni, J., Ferreira, J.P., White W.B., Cannon, C.P., Mehta, C.R., Bakris, G.L., Zannad, F. Validation and Update the Thrombolysis in Myocardial Infarction Risk Score for Heart Failure in Diabetes in Patients with Recent Acute Coronary Syndrome: An Analysis of the EXAMINE Trial. Diabetes, Obesity and Metabolism. 09 September 2022. DOI: 10.1111/dom.14867. Article ID: DOM14867. Internal Article ID: 17513551. Can the authors provide any models adjusting for this in the analysis as a sensitivity analysis.

5. There is additionally significant competing risk of events in post-MI populations : Hui SK, Sharma A, Docherty K, McMurray JJV, Pitt B, Dickstein K, Pfeffer MA, Girerd N, Rossignol P, Ferreira JP, Zannad F. Non-Fatal Cardiovascular Events Preceding Sudden Cardiac Death in Patients with an Acute Myocardial Infarction Complicated by Heart Failure: Insights from the High-Risk Myocardial-Infarction Database. European Heart Journal: Acute Cardiovascular Care. 2021 Apr 8;10(2):127-131. doi: 10.1093/ehjacc/zuaa012. The authors did not appear to use any competing risk models to account for this – which should be updated

Reviewer #3 (Remarks to the Author):

Ma and colleagues have studied the associations of S100A8/A9 plasma concentrations with heart failure (HF) development in patients with acute myocardial infarction (AMI). The authors have also performed a Mendelian Randomization (MR) study to explore if high S100A8/A9 concentrations are ‚causally‘ related to this endpoint.

1) As a courtesy to their readers, the authors need to provide some background information on S100A8/A9 upfront, in the Introduction (e.g. expression pattern, functional role after MI; this info is only provided in the Discussion). Early on, the authors also need to explain how essentially 2 proteins (S100A8 and S100A9) can be measured with a single ELISA and how the intracellular proteins S100A8/A9 are secreted?

2) All Methods that have been shifted to the Supplement should be incorporated into the main manuscript (or all Methods into the Supplement). I do not see how the methods described in the main manuscript are less important than the methods presented in the Supplement. Then, the reader does not have to constantly switch between the Manuscript and the Supplement.

3) The whole Ms. focuses on HF events, however, as stated on page 5, the authors really studied a combined endpoint of HF and cardiac death. This is not the same and needs to be corrected throughout. Cardiac death is not only related to HF. After discharge, “HF events were confirmed using medical records and/or contacting each patient/relative individually” (lines 126-127). All HF events need to be adjudicated; information provided by „patients“ or „relatives“ is not reliable.

4) Authors should provide a Figure illustrating S100A8/A9 plasma concentrations in relation to Killip class (I vs. II vs. III) on admission, and provide a Table illustrating S100A8/A9 plasma concentrations in relation to Clinical characteristics (as done in Table 1).

5) The Kaplan-Meier curves in Figs S11 and S15 use rather unusual cut-points to stratify the patient populations. The cut-points artificially magnify the prognostic power of S100A8/A9. Stratification according to quartiles would be more appropriate.

6) Infarct size is an important and established predictor of HF development after AMI (PMID: 27056772). The authors have adjusted for troponin I, measured only once on admission, which is a poor predictor of infarct size. This is a significant limitation.

7) Also, the authors repeatedly state that there are no established predictors for HF events after MI. This is not quite true; LVEF and (NT-pro)BNP measured before discharge are guideline-recommended tools for assessing that risk. Instead, the authors have adjusted for LVEF at baseline; how was this reliably done in the routine setting in patients sent for urgent primary PCI?

8) Authors need to better explain their “S100A8/A9” genetic score. Did they only choose “cis-pQTL on either side of the S100A9 gene” (Fig. S17). Why not cis-pQTL on either side of the S100A8 gene? Did all cis-pQTL predict elevated S100A8/A9 plasma concentrations (what about cis-pQTL predicting decreased S100A8/A9 plasma concentrations)? Authors should provide a scatter plot illustrating the association between their MR score and S100A8/A9 plasma concentrations.

Lines 95-66, line 135: what are “genetic instruments”?

Line 106: AMI patients are called ACS patients in Fig. S1. What is it?

Line 100: what are “early-onset AMI patients”?

Lines 238-240: Supplemental Figures 4, 5 do not show „unsupervised clustering“ data.

Line 239: n=2 contradicts Fig. S2 (3 x n=10).

Lines 243-244: „disease course“ (implying serial measurements?) is not studied.

Lines 252-253: While the basic science literature indicates that neutrophils are an important source of S100A8/A9, the authors cannot conclude that this is the case also in their patients (they only studied RNA and only inflammatory cells, leaving the possibility that other cells types contributed as well). Line 330 also needs to be rewritten accordingly.

Lines 253-254: it is not true that S100A8/A9/A12 was „exclusively produced“ by neutrophils (Figure S7, Fig. 2F).

Line 265: hazard ratio for what?

Lines 271-273: S100A12 was not associated with the combined endpoint also in univariate analysis.

Lines 340-341: „the prognostic role of S100A8/A9 in AMI remained unestablished“. Not true, Marinkovic et al. (Ref. 15) have reported that plasma S100A8/A9 during the acute post-ischemic phase is correlated with hospitalization for heart failure and cardiac function at 1 year after ACS. This limits the ‘novelty’ of their biomarker observations.

Line 350: what is „The triangulation of evidence“?

Lines 421-422: why „young“ AMI patients?

Reviewer #4 (Remarks to the Author):

Summary:

The authors perform a study aiming to identify proteins casual of increased risk of heart failure (HF) in patients with acute myocardial infarction (AMI). To do this, they identify candidate proteins for follow-up in a small cross-section cohort of 30 people: 10 AMI patients who developed HF during their hospital stay, 10 AMI patients who remained HF free at hospital discharge, and 10 healthy controls. After performing analyses of differential protein expression for ~1,000 proteins, they narrow down to two candidate targets; SA100-A8/A9 and S100A12. In a much larger prospective study of 1,062 AMI patients, they found that only SA100-A8/A9 and not S100-A12 measured at admission were predictive of increased risk of heart failure over a median of 4.2 years of follow-up. They further show this association replicated in an independent cohort of 1,183 patients with early onset AMI, and perform mediation analysis using a genetic score constructed from cis-pQTLs for S100-A9 identified in the deCODE pQTL mapping study, finding evidence suggesting that SA100-A8/A9 is a causal risk factor of heart failure development in AMI patients.

Major comments:

I have several comments related to the causal inference:

(1) First, I would argue that using Mendelian Randomization to describe the analysis performed could be misleading, and instead should be described instead as mediation analysis. This is an alternative framework for inferring causality that is quite different to most Mendelian Randomization methods readers would be more familiar with.

(2) I would strongly recommend the authors redo their mediation analysis with the medflex R package instead of the mediation R package. A major pitfall of the mediation R package as applied to the described analysis here is that by default it will be performing a counterfactual analysis of people with value of 0 for their S100A8/A9 genetic score to people with value of 1 for their S100A8/A9 genetic score. In contrast, the medflex package will perform counterfactual analysis over the full distribution of the S100A8/A9 genetic score, and as an added benefit, will also provide the estimated causal effect on a biologically meaningful scale (e.g. log OR per SD increase)

(3) While cis-pQTLs provide a strong prior on the genetic score influencing HF risk only through S100A8/A9, false positive causal inferences are still possible e.g. due to horizontal pleiotropy or impacts of genetic variation on protein binding efficacy (see Zheng et al. Nature Genetics 2020). Evidence for a causal effect could be strengthened by adding colocalization analyses to assess whether the pQTL signal and SNP-to-HF association arise from one (or more) shared causal variant(s).

(4) It would also be useful to see whether the causal effect of S100A8/A9 on risk of heart failure is significant using the more conventional two-sample Mendelian Randomization analysis, i.e. by overlaying the selected cis-pQTLs with GWAS summary statistics for Heart Failure. (E.g. Levin et al. Nature

Communications 2022 have published GWAS summary statistics for Heart Failure in the GWAS Catalog in a cohort included ~260,000 participants of East Asian ancestries). The R package TwoSampleMR could be used to do this analysis.

Following comment 4 above, it would be also interesting to see the results of two-sample Mendelian Randomization using a large Heart Failure GWAS to see whether the causal effect observed in patients with acute MI (assuming its robust) extends more generally to heart failure more broadly.

Regarding the identity of the target S100A8/A9, can the authors clarify whether this measurement is a protein complex of the A8 and A9 proteins? Or is it the case that the protein assays cannot easily distinguish between the two proteins?

Further to that comment, I note that the deCODE proteomics platform also includes a measurement for S100A8/A9 as well as S100A9 alone, but that the S100A8/A9 measurement did not have any cis (or trans) pQTLs.

I do not have the relevant expertise to critique the single-cell transcriptomics analysis, but I'm somewhat concerned by the very small sample size here (N=6). It's also not clear to me that these experiments meaningfully add to the conclusions of the paper, so I would suggest cutting these analyses.

I think it's also worth noting in the discussion that even for the initial differential protein expression scan the small sample size was quite small (N=30), so there may be other causal proteins that are being missed due to low detection power in the initial screening step.

It's somewhat unfortunate that so much of the space taken by the Main Figures are dedicated to analyses in the very small cross-sectional cohort, when the most impactful results come from the rest of the analyses and are shown mostly in the Supplemental Figures. In particular, I would suggest including a main Figure that shows (1) the significant differences in S100A8/A9 levels between HF and non-HF cases in both the discovery and validation cohorts, (2) the relationship between the genetic score for S100A8/A9 and measured S100A8/A9 in the healthy controls. E.g. A figure combining the current Figure 3 with Supplemental Figures S8, S11-16, additionally showing the validity of the genetic score for S100A8/A9 .

Another point that needs expanding on in the discussion is the differences between the validation cohort and the discovery cohort. The validation cohort participants are unusually young for acute MI patients. Is it possible that these patients have very different characteristics in terms of heart failure risk to patients whose age of onset of acute MI are more typical, and in turn, possible that the observed causal effect of S100A8/A9 on HF risk only applies to people with early-onset MI?

The follow-up time for both the discovery and validation cohorts extends beyond 2020, presenting the possibility of confounding from the COVID19 pandemic on heart failure risk (e.g. due to differences in treatment quality/availability during periods of large amounts of COVID19 cases, differences in lifestyle/behaviour due to lockdowns, or differences arising due to SARS-CoV2 infection or exposure). It would be great to see a sensitivity analysis of the S100A8/A9 associations with heart failure risk when truncating the follow-up time to the start of 2020.

Minor comments:

Can the authors clarify whether the study termination time (May 31st 2021) was the maximum follow-up for both the discovery and validation cohorts?

Supplementary Figure 8 and Supplementary Figure 13 would be clearer if the median and interquartile range were shown above the scatterplot points.

Supplementary Figure 2 does not match the methods and results text: Supplementary Figure 2 shows 30 participants had scRNA-seq data, whereas the text states only 6 participants had these data.

Reviewer #1:

Ma et al. used protein array-based serum proteomics and single cell transcriptomics to select candidate biomarkers for post-acute myocardial Infarction Heart Failure (Post-AMI HF), then used Mendelian Randomization study to confirm the causal effects of S100A8/A9 in Post-AMI HF. The results may be interesting for predicting Post-AMI HF, however, there are several issues in the manuscript, as mentioned below, that should be clarified.

Reply, We greatly appreciate the reviewer's constructive suggestions that aimed to improve the quality of our study. We have specifically addressed each concern below.

1. In this manuscript, the authors used protein array-based technology to detect the levels of 1000 proteins in three different groups. The authors detected both S100A8 and S100 A8/A9 as differential proteins across the three groups (Supplemental table 3). Though S100A8 preferentially exists as a heterodimer or heterotetramer with S100A9 as calprotectin (S100A8/A9), how about the level of S100A9 in the three groups? Could the level of S100A9 remain the same in the three groups, while the level of S100A8 increased significantly in HF group? Mass spectrometry-based proteomic technology, such as MRM-based targeted proteomic method, may provide more information about the level of S100A9 in the three groups.

Reply 1, We did not describe it clearly. The level of S100A9 in the three groups was not detected, because it is not included in the panel of 1000 protein array-based technology. As Reviewer suggested, it is indeed necessary to evaluate whether and how S100A9 level is correlated with post-AMI HF. In order to answer this question, we simultaneously detect the levels of S100A9 and S100A8/A9 in same sample from three groups using ELISA. The result shows that the level of S100A9 is highly correlated with the level of S100A8/A9, and the increase of S100A9 level is also associated with post-AMI HF (*Supplemental Figure 19*). S100A9 levels is far below S100A8/A9 levels, which is consistent with previous report¹. It may be because S100A8 and S100A9 monomers preferentially form stable heterodimers as their basic form.

Supplemental Figure 19. Correlation between serum S100A8/A9 levels and S100A9 levels

(A) Scatter plot of the correlation between S100A8/A9 levels and S100A9 levels. Distribution of S100A9 (B) and S100A8/A9 (C) in HCs (n=24), AMI patients without (n=24) and with HF (n=24).

2. In both discovery cohort and validation cohort, the authors detected the level of S100A8/A9 using ESILA and found that the level of S100A8/A9 was increased in HF patients compared to no-HF patients (Supplemental Figure 8 and Figure 13). I am wondering whether higher level of S100A8/A9 definitely indicates a higher risk of Post-AMI HF? or whether there is a

concentration range of S100A8/A9 which could be as a reference for a higher risk of Post-AMI HF?

Reply 2, Kaplan–Meier curves illustrated that patients in the higher-risk categories stratified by the quartile of S100A8/A9 exhibited a higher risk of post-AMI HF events in both cohorts. The cutoff values of S100A8/A9 in the high-risk group were 5,059 ng/mL and 4,877 ng/mL in the discovery and validation cohorts, respectively, suggesting that S100A8/A9 level exceeding 5,000 ng/mL might be a reference for a higher risk of HF (*Revised Figure 3*). (*Revised position: Page 10, Lines 4-9, in Result section*)

Revised Figure 3 Incidence of HF events according to S100A8/A9 levels

Kaplan–Meier curves illustrate the timing of heart failure (HF) events in the four strata of S100A8/A9 levels. The quartile of S100A8/A9 was used to classify patients into four risk categories, which were low, low- intermediate, intermediate-high, and high risk in the discovery cohort (A) and validation cohort (B).

A

B

3. The authors states that “If a biomarker is causally associated with HF, it not only helps the prediction of HF, but also serves as the target of intervention”(line 69-71), could S100A8/A9 be used as target for intervention of Post-AMI HF and how S100A8/A9 could be used as intervention of post-AMI HF?

Reply 3, We apologized for this imprecise description and revised into “If a biomarker is causal risk factor for HF, it not only predicts the occurrence of HF, but also is involved in the development of HF”.

Bench work has identified S100A8/A9 as a driver of ischemia and reperfusion injury (*Detailed information were shown in the Discussion Page 15, Lines 10-25*). In present study, we prove the causal effect of S100A8/A9 on post-AMI HF in human. Both human and experimental study indicate that S100A8/A9 is a promising therapeutic target for post-AMI HF. Administration with ABR238901(an orally active and potent S100A8/A9 blocker) or a S100a9 neutralizing antibody, could prevent cardiac injury and HF post AMI in animal model ^{2,3}. Thus, targeting the S100A8/A9 with respect to HF prevention, is worth exploring (*Revised position: Page 16, Lines 21-25 in Discussion section*).

Reviewer #2:

I am reviewing for the first time the article by Ma and colleagues which aims to identify independent markers of inflammation in a post-MI population. The authors identified S100A8/A9 as a potential causative marker associated with HF in post-MI populations. This is an overall well-designed study with unique insights. I have the following comments:

Reply, We thank the reviewer for the positive comments. We have specifically addressed each concern below.

1. Did heart failure outcomes include future assessment of LVEF? This appears to be a baseline covariate but not recorded in the future

Reply 1, We fully agree with Reviewer that LVEF is important indicator for HF. Here, we did not record LVEF after discharge based two reasons: i) Clinical outcomes include in-hospital HF and long-term HF in our study. Adjudication of HF events did not need the assessment of LVEF. Definitions of HF: In-hospital HF included new HF onset, worsening HF, cardiogenic shock diagnosis, and in-hospital death due to HF or cardiogenic shock. Long-term HF included HF progression resulting in rehospitalization and/or death due to HF after the initial discharge (*Supplemental Table 3*). ii) It is hard to perform echocardiogram in the total of 2105 patients during median follow-up times 4.2 years (IQR, 1.7–5.1) and 2.9 years (IQR, 1.6–4.2) in discovery and validation cohorts.

2. The incremental improvement in c-statistic is limited in the validation cohorts – i.e. from 0.7 to 0.73 from reference model to model that includes S100A8/A9. Can the authors corroborate whether they feel this is a significant improvement? If so, then why?

Reply 2, We determined the significance of improvement in c-statistic based on P-value less than 0.05. We tried but failed to find any literature or standard indicating the minimum value of incremental improvement in C-statistic when it was considered to be significant. We performed multiple analysis to evaluate the improvement of S100A8/A9 on top of reference model. In new version, we recalculated the Δ C-statistic and NRI since long-term HF events (exclusion of non-HF death) and reference model (addition of infarction area calculated by AUC of CK-MB) have changed.

Firstly, S100A8/A9 increased C-statistic significantly when added to the reference model in discovery cohort (Δ C-statistic: 0.04 [95% CI: 0.02–0.05], $P < 0.001$) and validation cohort (Δ C-statistic: 0.03 [95% CI: 0.01–0.05], $P < 0.001$). Secondly, NRI analysis showed that the addition of S100A8/A9 to the reference model improved the classification of events in discover cohort (NRI: 0.23 [95% CI: 0.12–0.32], $P < 0.001$) and validation cohort (NRI: 0.10 [95% CI: 0.03–0.23], $P < 0.001$) (*Revised Table 2*). Thirdly, we further calculated a clinical “net benefit” for S100A8/A9 in comparison to reference model using decision curve analysis. We set up the decision threshold base on of the incidence of HF events in two cohorts. Compared with reference model, the model including S100A8/A9 would identify ~34 or 19 additional cases on average, without identifying any additional false positive, in a population of 1000 patients (*Supplemental Figure 13*). (*Revised position: Page 8, Lines 16-25, Page 9, Lines 1-4 and 15-18, in Result section*). Corresponding method of decision curve analysis was added into Supplemental material (*Revised position: Page 13, Lines 8-10*).

Revised Table 2. Discrimination/reclassification improvement by S100A8/A9 and S100A12 for HF risk prediction

	C-statistic (95% CI)	Δ C-statistic (95% CI)	P-value	NRI (95% CI)	P-value
Discovery cohort					
S100A8/A9	0.65 (0.62–0.68)				
Reference model	0.70 (0.67–0.73)	Reference		Reference	
				0.23 (0.12–0.32)	
+S100A8/A9	0.74 (0.71–0.77)	0.04 (0.02–0.05)	<0.001	Non-Events: 0.14 (0.09–0.23) Events: 0.09 (0.01–0.12)	<0.001
Validation cohort					
S100A8/A9	0.70 (0.66–0.74)				
Reference model	0.79 (0.76–0.82)	Reference		Reference	
				0.10 (0.03–0.23)	
+S100A8/A9	0.82 (0.78–0.85)	0.03 (0.01–0.05)	0.001	Non-Events: 0.052 (0.006–0.100) Events: 0.052 (0.002–0.154)	0.009

The reference model included age, sex, systolic blood pressure, Killip classification at admission, fasting glucose, creatinine, left main artery disease, neutrophil count, cardiac troponin I, B-type natriuretic peptide, c-reactive protein, left ventricular ejection fraction at admission and estimated infarct size (CK-MB AUC₀₋₇₂). Net reclassification improvement (NRI) was assessed using the two category-based NRIs using 10% and 30% as HF-event cut-offs to define patient subgroups at low, intermediate, or high risk.

Supplemental Figure 13. Decision curve analysis of combined S100A8/A9 and reference model vs. reference model alone for the HF events

The reference models included age; sex; systolic blood pressure (SBP); Killip classification at admission; fasting glucose; creatinine; left main artery disease; neutrophil count; cardiac troponin I, B-type natriuretic peptide, and C-reactive protein levels; left ventricular ejection fraction at admission; and estimated infarct size (CK-MB AUC₀₋₇₂).

3. Use of mineralocorticoid receptor antagonist (MRA) such as spironolactone are evidence based in a post MI population. Could needs to be included in baseline tables and also in the baseline model as a sensitivity analysis

Reply 3, As Reviewer suggested, the use of mineralocorticoid receptor antagonist (MRA) was added into the baseline tables and was similar between HF and no-HF patients (*Revised Table 1*).

As only potential confounders ($P < 0.05$ in univariate analysis) were included in multivariate analysis, MRA (HR: 0.88[95%CI 0.64-1.23], $P=0.455$) was not included in multivariate adjustment.

Sensitivity analysis indicated that high S100A8/A9 levels remained associated with HF after the adjustment in the patients with MRA treatment (Discovery cohort: HR: 2.70 [95%CI: 1.66-4.37], $P < 0.001$; validation cohort: HR: 8.78 [95%CI: 2.06-37.50], $P = 0.003$) and without MRA treatment (Discovery cohort: HR: 1.93 [95%CI: 1.65-2.25], $P < 0.001$; Validation cohort: HR: 2.15 [95%CI: 1.78-2.61], $P < 0.001$) MRA treatment.

Revised Table 1 Clinical characteristics according to HF events in discovery and validation cohorts

Variables	Discovery cohort (n=1062)			Validation cohort (n=1043)		
	HF events (n=296)	No-HF events (n=766)	P- value	HF events (n=192)	No-HF events (n=851)	P- value
MRA	42 (14.2)	133 (17.4)	0.376	15 (7.8)	60 (7.1)	0.712

MRA: mineralocorticoid receptor antagonist

4. In post-MI populations, urine albumin-creatinine ratio is a very strong prognostic marker of adverse HF outcomes. See Razaghizad, A., Sharma, A., Ni, J., Ferreira, J.P., White W.B., Cannon, C.P., Mehta, C.R., Bakris, G.L., Zannad, F. Validation and Update the Thrombolysis in Myocardial Infarction Risk Score for Heart Failure in Diabetes in Patients with Recent Acute Coronary Syndrome: An Analysis of the EXAMINE Trial. Diabetes, Obesity and Metabolism. 09 September 2022. DOI: 10.1111/dom.14867. Article ID: DOM14867. Internal Article ID: 17513551. Can the authors provide any models adjusting for this in the analysis as a sensitivity analysis.

Reply 4, We very much regret to not provide this analysis, because the 90% missing of urine albumin-creatinine ratio in two cohorts.

TRS-HF_{DM} score is developed and validated to predict HF in patients with type 2 diabetes (T2DM). The urine albumin-creatinine ratio was included in TRS-HF_{DM} score and account for 1 point. In Faiez Zannad's study, participates were eligible for inclusion if they had T2DM and an ACS (i.e. acute myocardial infarction, unstable angina) in the past 15-90 days. The 67% and 80% patients has no history of T2DM in our discovery and validation cohort. It is not clear whether urine albumin-creatinine ratio is a strong prognostic marker of HF outcomes in population without T2DM.

5. There is additionally significant competing risk of events in post-MI populations : Hui SK, Sharma A, Docherty K, McMurray JJV, Pitt B, Dickstein K, Pfeffer MA, Girerd N, Rossignol P, Ferreira JP, Zannad F. Non-Fatal Cardiovascular Events Preceding Sudden Cardiac Death in Patients with an Acute Myocardial Infarction Complicated by Heart Failure: Insights from the High-Risk Myocardial-Infarction Database. European Heart Journal: Acute Cardiovascular Care. 2021 Apr 8;10(2):127-131. doi: 10.1093/ehjacc/zuaa012. The authors did not appear to use any competing risk models to account for this – which should be updated

Reply 5, We appreciated the Reviewer's constructive advice. The causes of death after AMI in two

cohorts were listed (*Supplemental Table 2*). The “insufficient evidence to determine death from HF” and “non-cardiac death” are competing risk of events. After controlling for competitive risk events, Fine-Gray (FG) model demonstrated that S100A8/A9 was associated with a higher incidence of post-AMI HF in discovery cohort (HR: 2.03 [95% CI: 1.76-2.34], P < 0.001) and validation cohort (HR: 2.14 [95% CI: 1.76-2.60], P < 0.001). (*Revised position: page 8, line 5-8 and page 9, line 9-12 in Result section*). Corresponding method of FG was added into Supplemental material (*Revised position: Page 12, Lines 16-22*).

Supplemental Table 2. Occurrence of adverse events in the discovery cohort and validation cohort during the follow-up period

	HF events					Cardiac death	Non-cardiac death
	In-hospital		Long-term			Insufficient evidence to determine death from HF	Neoplasms/ Traffic accident
	new HF onset	CS	Death due to HF/CS	HF progression resulting in re-hospitalization	Death due to HF		
Discovery (n=1062)	66	35	17	112	66	18	3
Validation (n=1043)	70	30	10	66	16	2	1

Reviewer #3:

Ma and colleagues have studied the associations of S100A8/A9 plasma concentrations with heart failure (HF) development in patients with acute myocardial infarction (AMI). The authors have also performed a Mendelian Randomization (MR) study to explore if high S100A8/A9 concentrations are causally related to this endpoint.

Reply, We greatly appreciate the reviewer's constructive suggestions that aimed to improve the quality of our study. We have specifically addressed each concern below.

1. As a courtesy to their readers, the authors need to provide some background information on S100A8/A9 upfront, in the Introduction (e.g. expression pattern, functional role after MI; this info is only provided in the Discussion). Early on, the authors also need to explain how essentially 2 proteins (S100A8 and S100A9) can be measured with a single ELISA and how the intracellular proteins S100A8/A9 are secreted?

Reply 1, Thanks for Reviewer's suggestion. We revised the Introduction as follows:

This study aimed to identify the biomarkers associated with HF in patients with AMI. We selected S100A8/A9 as a potential biomarker, using integrated proteomic and single-cell transcriptomic analyses. S100A8 and S100A9, as endogenous alarmins, are constitutively expressed in myeloid cells and stored as granules that are ready to be released in response to infectious or bacteria-free inflammation. They exist in several forms but preferentially form a heterodimeric complex of S100A8/A9, which is necessary for their biological effects⁵. The secretion of S100A8/A9 is partly dependent on the reactive oxygen species (ROS) and potassium Efflux. S100A8/A9 is also released during NETosis. In the MI setting, both excessive ROS and NETosis are conducive to S100A8/A9 release into the heart and circulation.⁶ Consequently, we prospectively validated the predictive values of S100A8/A9 in two independent cohorts and evaluated the causal relationship between S100A8/A9 and post-AMI HF. (*Revised position: Page 4, Lines 4-15 in Introduction section*).

We detect the heterodimeric complex of S100A8/A9 using ELISA kit (S100A8/9: R&D Systems Europe, Abingdon, Oxford, UK). The assay employed a quantitative sandwich enzyme immunoassay technique. Briefly, a monoclonal antibody specific to the human S100A8/S100A9 heterodimer was pre-coated onto a microplate. Subsequently, the samples were pipetted into the wells. After washing away the unbound substances, an enzyme-linked monoclonal antibody specific for the human S100A8/S100A9 heterodimer was added to each well. All steps strictly followed the operation procedure of ELISA protocol. (*Revised position: Page 8, Lines 21-25 and Page 9, Lines 1-2 in Supplemental Material*).

2. All methods that have been shifted to the Supplement should be incorporated into the main manuscript (or all Methods into the Supplement). I do not see how the methods described in the main manuscript are less important than the methods presented in the Supplement. Then, the reader does not have to constantly switch between the Manuscript and the Supplement.

Reply 2, As Reviewer suggested, we moved all Methods to Supplemental Material. This decidedly increases the readability of our manuscript.

3. The whole Ms. focuses on HF events, however, as stated on page 5, the authors really studied a combined endpoint of HF and cardiac death. This is not the same and needs to be corrected

throughout. Cardiac death is not only related to HF. After discharge, “HF events were confirmed using medical records and/or contacting each patient/relative individually” (lines 126-127). All HF events need to be adjudicated; information provided by “patients” or “relatives” is not reliable.

Reply 3, We are grateful for Reviewer's constructive advice. We redefined the cardiac death. In-hospital death due to HF or cardiogenic shock was defined as a death with clinical, radiologic, or postmortem evidence of HF, in the absence of in-hospital acute ischemic event. Long-term death due to HF was defined as death in the context of clinically worsening symptoms and/or signs of HF with no other apparent cause, death as a consequence of a surgical procedure to treat HF, or death after referral to hospice for HF (*Supplemental Table 3*). All HF events were adjudicated by the consensus of two experienced cardiologists who were blinded to study results by review of outpatient clinics or hospitalization records and telephone interview. In current version, 118, 110 in-hospital and 178, 82 long-term HF events were recorded in discovery and validation cohort (*Supplemental Table 2*). (*Revised position: Page 5, Lines 3-24 in Supplemental Material*).

Supplementary Table 3 Definitions of the HF events

Outcome		Definition
in-hospital	new HF onset	The new onset of signs and symptoms of HF were clinical manifestations such as dyspnea, orthopnea, peripheral edema, jugular vein dilatation, a third heart sound (S3), and lung rale. ⁴
	worsening heart failure	Killip class II progressed to III or IV, Killip class III progressed to IV
	cardiogenic shock	systolic blood pressure ≤ 90 mm Hg for > 30 min after exclusion of hypovolaemia, with clinical evidence of hypoperfusion, inotrope dependence, or mechanical left ventricular support to correct the issue.
	Death due to heart failure or cardiogenic shock	Death due to HF or cardiogenic shock was defined as a death with clinical, radiologic, or postmortem evidence of HF, in the absence of in-hospital acute ischemic event.
Long-term	HF progression resulting in re-hospitalization	a hospital readmission for which HF was the primary reason. It was specifically defined as an event meeting all of the following criteria: (1) admission to the hospital for at least 24 hours; (2) objective evidence of new or worsening HF (e.g., orthopnea, jugular venous distension, pulmonary basilar crackles, etc.); and (3) intensification of HF therapy (e.g., initiation of intravenous diuretics or inotropes).
	Death due to heart failure	Death due to heart failure was defined as death in the context of clinically worsening symptoms and/or signs of heart failure with no other apparent cause, death as a consequence of a surgical procedure to treat heart failure, or death after referral to hospice for heart failure.

Supplemental Table 2. Occurrence of adverse events in the discovery cohort and validation cohort during the follow-up period

	HF events					Cardiac death	Non-cardiac death
	In-hospital			Long-term		Insufficient evidence to determine death from HF	Neoplasms/ Traffic accident
	new HF onset	CS	Death due to HF/CS	HF progression resulting in re-hospitalization	Death due to HF		
Discovery (n=1062)	66	35	17	112	66	18	3
Validation (n=1043)	70	30	10	66	16	2	1

4. Authors should provide a Figure illustrating S100A8/A9 plasma concentrations in relation

to Killip class (I vs. II vs. III) on admission, and provide a Table illustrating S100A8/A9 plasma concentrations in relation to Clinical characteristics (as done in Table 1).

Reply 4, As Reviewer suggested, Killip class is a specialized indicator for cardiac function in patients with AMI. We observed that the S100A8/A9 concentrations were higher in Killip class II and III patients than in Killip class I patients in the validation cohort (**Supplemental Figure 14**). (*Revised position: Page 9, lines 19-22 in Result section*). Associations between baseline characteristics of the two cohorts and quartiles of S100A8/A9 are shown (*Supplemental Table 13*). Neutrophil counts, cTnI, LVEF, infarction size estimated by AUC of CK-MB₀₋₇₂ progressively increased among S100A8/A9 quartiles in both cohorts. (*Revised position: Page 10, Lines 9-10 in Result section*).

Supplemental Figure 14. S100A8/A9 plasma concentrations in Killip class on admission in discovery and validation cohorts

Supplemental Table 13 Baseline characteristics of the study population grouped by the quartiles of S100A8/A9

Variables	Discovery cohort (n=1062)					Validation cohort (n=1043)				
	≤ 2346 (n = 265)	2346-3641 (n = 266)	3641-5069 (n= 266)	>5069 (n = 265)	P for trend	≤ 1905 (n =261)	1905-3657 (n = 261)	3657-4877 (n= 261)	>4877 (n = 260)	P for trend
Demographics										
Age (years)	62.0 (54.0-69.0)	60.0 (50.0-68.0)	59.0 (52.0-67.3)	55.0 (48.0-64.0)	<0.001	39.0 (36.0-41.0)	39.0 (36.0-43.0)	40.0 (37.0-43.0)	40.0 (37.0-42.0)	0.196
Male sex	207 (78.1)	199 (74.8)	221 (83.1)	224 (84.5)	0.012	238 (91.2)	233 (89.3)	2843 (93.1)	237 (91.2)	0.365
SBP (mm Hg)	120.0 (107.0-132.0)	120.0 (110.0-135.0)	120.0 (110.0-137.0)	120.0 (107.0-135.0)	0.556	127.0 (117.5-136.0)	125.0 (119.0-132.0)	116.0 (122.0-131.0)	123.0 (116.0-130.8)	0.010
DBP (mm Hg)	70.0 (65.0-80.0)	72.0 (70.0-80.0)	73.0 (67.8-84.0)	75.0 (69.0-83.0)	0.037	79.0 (70.0-86.0)	80.0 (70.0-83.5)	78.0 (70.0-82.0)	79.0 (70.0-81.9)	0.430
Current smoking	149 (56.2)	163 (61.3)	162 (60.9)	178 (67.2)	0.015	184 (70.5)	161 (61.7)	173 (66.3)	162 (62.3)	0.130
Killip classification					0.332					0.001
I	225 (84.9)	218 (82.0)	219 (82.3)	217 (81.9)		249 (95.4)	252 (96.6)	235 (90.0)	235 (90.4)	
II	38 (14.3)	40 (15.0)	43 (16.2)	42 (15.8)		11 (4.2)	8 (3.1)	20 (7.7)	19 (7.3)	
III	2 (0.8)	8 (3.0)	4 (1.5)	6 (2.3)		1 (0.4)	1 (0.4)	6 (2.3)	6 (2.3)	
Medical history										
Hypertension	156 (58.9)	157 (59.0)	152 (57.1)	156 (58.9)	0.889	122 (46.7)	131 (50.2)	124 (47.5)	136 (52.3)	0.312
Hyperlipidemia	167 (63.0)	164 (61.7)	170 (63.9)	194 (73.2)	0.012	121 (46.4)	119 (45.6)	122 (46.7)	120 (46.2)	0.969
Diabetes mellitus	96 (36.2)	91 (34.2)	89 (33.5)	79 (29.8)	0.123	51 (19.5)	53 (20.3)	54 (20.7)	55 (21.2)	0.640
CAD	80 (30.2)	65 (24.4)	64 (24.1)	49 (18.5)	0.003	56 (21.5)	58 (22.2)	52 (19.9)	43 (16.5)	0.125
Biochemical										
Neutrophil counts (×10 ⁹ /L)	5.4 (4.1-7.1)	6.1 (4.4-8.3)	6.7 (5.1-8.6)	7.2 (5.3-9.3)	<0.001	5.2 (4.2-7.1)	5.7 (4.6-7.1)	5.9 (4.7-8.1)	6.8 (5.1-8.7)	<0.001
HDL cholesterol (mmol/L)	1.0 (0.9-1.2)	1.0 (0.9-1.2)	1.1 (0.9-1.2)	1.0 (0.9-1.2)	0.773	0.9 (0.8-1.1)	1.0 (0.8-1.1)	0.9 (0.8-1.1)	0.9 (0.8-1.1)	0.299
LDL cholesterol (mmol/L)	2.7 (2.3-3.4)	2.9 (2.3-3.4)	2.9 (2.2-3.5)	2.9 (2.2-3.4)	0.779	2.3 (1.8-3.2)	2.3 (1.8-3.0)	2.3 (1.7-2.9)	2.2 (1.8-3.0)	0.200
Fasting glucose (mmol/L)	6.7 (5.6-9.3)	6.4 (5.5-9.0)	6.7 (5.6-9.1)	7.2 (5.6-9.7)	0.543	5.6 (5.1-6.9)	5.4 (5.0-6.6)	5.5 (5.0-6.3)	5.4 (5.0-6.7)	0.604
Creatinine (μmol/L)	72.9 (63.2-85.4)	72.6 (61.9-84.2)	73.3 (62.4-83.3)	74.5 (62.8-85.3)	0.520	71.3 (63.0-78.6)	69.5 (64.6-78.8)	72.0 (63.8-80.0)	71.7 (64.8-81.5)	0.740
Biomarkers										
cTnI (ng/mL)	0.9 (0.1-5.4)	1.3 (0.1-9.5)	2.1 (0.2-14.3)	1.9 (0.2-11.5)	0.024	1.2 (0.2-6.5)	1.2 (0.2-4.6)	1.2 (0.3-6.6)	0.8 (0.1-9.1)	0.001
BNP (pg/mL)	101.0 (40.0-234.0)	100.0 (48.5-225.1)	110.0 (40.5-262.0)	127.0 (47.5-256.0)	0.581	78.6 (28.0-165.0)	83 (27.0-189.0)	75.0 (29.0-191.4)	96.6 (32.3-221.3)	0.003
hs-CRP (mg/L)	5.0 (1.8-15.8)	5.4 (1.9-19.9)	5.8 (2.2-17.5)	8.0 (2.7-21.9)	0.083	4.9 (1.6-15.4)	4.1 (1.5-11.7)	4.3 (1.7-12.9)	5.7 (1.9-15.5)	0.187
Overall lesion profiles										

Multi-vessel disease					0.243					0.145
Left main artery disease	10 (3.8)	12 (4.5)	8 (3.0)	2 (0.8)		6 (2.3)	14 (5.4)	14 (5.4)	6 (2.3)	
2-vessel disease	82 (30.9)	80 (30.1)	73 (27.4)	76 (28.7)		60 (23.0)	47 (18.0)	38 (14.6)	47 (18.1)	
3-vessel disease	38 (14.3)	29 (10.9)	37 (13.9)	33 (12.5)		59 (22.6)	62 (23.8)	48 (18.4)	57 (21.9)	
Echocardiography										
Admission LVEF (%)	55.0 (50.0-60.0)	55.0 (49.0-58.0)	53.0 (47.0-58.0)	53.0 (47.0-58.0)	<0.001	56.0 (52.0-60.0)	55.0 (50.0-60.0)	55.0 (50.0-60.0)	55.0 (47.0-58.0)	<0.001
Infarct size										
CK-MB-based estimation	4445.1 (1564.4-6407.6)	4863.4 (1560.4-6921.1)	5277.7 (2752.1-7444.9)	5577.4 (2619.8-7845.5)	<0.001	4253.4 (2073.6-6108.0)	4566.5 (2432.1-60401.0)	4357.6 (2380.1-6178.0)	4534.6 (2522.7-7422.7)	0.006
Medication at discharge										
Aspirin	264 (99.6)	257 (96.6)	261 (98.1)	256 (96.6)	0.629	257 (98.5)	254 (97.3)	248 (95.0)	251 (96.5)	0.095
P2Y12 receptor Inhibitor	265 (100.0)	261 (98.1)	262 (98.5)	257 (97.0)	\	253 (96.9)	244 (93.5)	230 (88.1)	237 (91.2)	0.002
Statin	261 (98.5)	250 (94.0)	256 (96.2)	252 (95.1)	0.899	253 (96.9)	248 (95.0)	245 (93.9)	244 (93.8)	0.087
ACEI or ARB	156 (58.9)	145 (54.5)	166 (62.4)	139 (52.5)	0.641	114 (43.7)	122 (46.7)	104 (39.8)	109 (41.9)	0.375
Beta-blockers	201 (75.8)	192 (72.2)	191 (71.8)	199 (75.1)	0.744	176 (67.4)	183 (70.1)	172 (65.9)	198 (76.2)	0.085
MRA	41 (15.4)	50 (18.9)	48 (18.0)	36 (13.6)	0.635	19 (5.7)	18 (9.7)	19 (6.3)	14 (6.4)	0.151

5. The Kaplan–Meier curves in Figs S11 and S15 use rather unusual cut-points to stratify the patient populations. The cut-points artificially magnify the prognostic power of S100A8/A9. Stratification according to quartiles would be more appropriate.

Reply 5, As Reviewer suggested, we used quartiles as cut-points to replot the Kaplan–Meier curves. Kaplan–Meier curves illustrated that patients at higher risk categories stratified by the quartile of S100A8/A9 have the higher risk of post-AMI HF events in both cohorts (*Revised Figure 3*). (*Revised position: Page 10, Lines 4-9 in Result section*).

Revised Figure 3 Incidence of HF events according to S100A8/A9 levels

Kaplan–Meier curves illustrate the timing of heart failure (HF) events in the four strata of S100A8/A9 levels. The quartile of S100A8/A9 was used to classify patients into four risk categories, which were low, low- intermediate, intermediate-high, and high risk in the discovery cohort (A) and validation cohort(B).

6. Infarct size is an important and established predictor of HF development after AMI (PMID: 27056772). The authors have adjusted for troponin I, measured only once on admission, which is a poor predictor of infarct size. This is a significant limitation.

Reply 6, We agree with Reviewer that only cTnI on admission are not sufficient to predict the infarct size. Kinetic profiles of biomarkers show CK-MB as the most accurate indicator in determining infarct size⁷. Infarct size was evaluated using the area under the curve (AUC) for the CK-MB enzyme over the first 72 h after admission (CK-MB AUC₀₋₇₂). AUC of CK-MB was calculated based on 5 values (at 0, 6, 24, 48, and 72 hours after enrollment) using the linear trapezoidal method (*Supplemental Figure 18*). Corresponding methods were provided in Supplemental Material (*Page 9, Lines 23~25 and Page 10, Lines 1~3*). CK-MB AUC₀₋₇₂ and cTnI on admission were simultaneously incorporated in multivariate adjustment model and reference model. In multivariate analysis, HF-S100A8/A9 association remained significant in discovery cohort (*Revised Figure 2B*, HR: 2.03 [95% CI: 1.77–2.33], P < 0.001) and validation cohort (*Revised Figure 2E*, HR: 2.15 [95% CI: 1.79–2.58], P < 0.001). (*Revised position: Page 7, Line 24 and Page 9, Line 8 in Result section*).

Supplemental Figure 18. Estimation of infarct size in discovery and validation cohorts

Measures of infarct size by 72-h creatine kinase-MB isoenzyme (A, B) release. CK-MB, creatine kinase-MB isoenzyme.

Revised Figure 2B and 2E. Hazard ratios for HF events in the discovery cohort.

Unadjusted and adjusted Hazard ratios for HF events from Cox proportional hazards regression analysis were shown for discovery (B) and validation (E) cohort. S100A8/A9 was measured on admission. Model 1: adjusted for age and sex; model 2: adjusted for model 1+systolic blood pressure, Killip classification at admission, fasting glucose, creatinine, left main artery disease; model 3: adjusted for model 2+neutrophil count, cardiac troponin I, B-type natriuretic peptide, c-reactive protein, left ventricular ejection fraction at admission and estimated infarct size (CK-MB AUC₀₋₇₂).

7. Also, the authors repeatedly state that there are no established predictors for HF events after MI. This is not quite true; LVEF and (NT-pro) BNP measured before discharge are guideline-recommended tools for assessing that risk. Instead, the authors have adjusted for LVEF at baseline; how was this reliably done in the routine setting in patients sent for urgent primary PCI?

Reply 7, We appreciated for Reviewer's correction. We turn down the wording in the paper. The statement “no appropriate biomarkers for early HF prediction” was uniformly modified as “the existing known biomarkers are insufficient for precise HF prediction in patients with AMI.” (*Revised position: Page 13, Lines 9-10 in Discussion section*).

Use of echocardiography helps to evaluate detailed hemodynamic information and suspected complication of myocardial ischemia/infarction. Guidelines for the ultrasonography recommend that patients with suspected ACS and AMI should undergo bedside cardiac ultrasound (Grade 1C)⁸. Moreover, point-of-care ultrasonography has been routinely used in Anzheng hospital which is the national clinical center for cardiovascular diseases. Therefore, we widely performed emergency bedside ultrasound examination before primary PCI to obtain baseline LVEF.

8. Authors need to better explain their “S100A8/A9” genetic score. Did they only choose “cis-pQTL on either side of the S100A9 gene” (Fig. S17). Why not cis-pQTL on either side of the S100A8 gene? Did all cis-pQTL predict elevated S100A8/A9 plasma concentrations (what about cis-pQTL predicting decreased S100A8/A9 plasma concentrations)? Authors should provide a scatter plot illustrating the association between their MR score and S100A8/A9 plasma concentrations.

Reply 8, To establish genetic instruments associated with S100A8/A9 level, we search for genome-

wide association studies (GWAS) summary statistics of plasma proteins for all 4,907 aptamers at <https://www.Decode.com/summarydata/>. However, we could not find GWAS summary statistics for S100A8/A9 or S100A8. Because S100A9 regulates S100A8/A9 complex functions through various mechanisms, including protecting S100A8 from degradation, and S100A9 levels were strongly correlated with S100A8/A9 levels in the plasma ($r=0.92$), we selected 50 pQTLs of S100A9 from the GWAS of 35,559 Icelanders. The 24 SNPs with $\beta > 0$ are associated with elevated S100A8/A9 concentrations, and 26 SNPs with $\beta < 0$ are associated with decreased S100A8/A9 concentrations (**Supplemental Table 14**). In our validation cohort, rs12033317 ($\beta = 0.33$) and rs12119788 ($\beta = 0.89$) were associated with increased S100A8/A9 levels, and rs1560832 ($\beta = -0.52$), rs3014874 ($\beta = -0.47$), rs3014875 ($\beta = -0.42$), and rs59961408 ($\beta = -0.16$) were associated with decreased S100A8/A9 levels, consistent with the directions in the GWAS summary statistics. (**Revised position: Page 10, Lines 16-25 and Page 11, lines 15-19 in Result section**).

We provided a scatter plot illustrating the positive association between the genetic score and S100A8/A9 plasma concentrations for validation cohort (**Supplemental Figure 16**) (**Revised position: Page 11, lines 19-21 in Result section**).

Supplemental Table 14. 50 S100A9 cis-pQTL and their association with S100A9 in the GWAS summary statistic

Chrom	Pos (GRCh38)	rsids	effectAllele	other Allele	Beta	SE	P-value	N	ImpMAF	MAF in Asian
chr1	153344712	NA	C	G	0.2869	0.1442	0.0466	35363	0.00111	0.00000
chr1	153345524	rs552917187	G	A	-0.1508	0.0463	0.0011	35363	0.00791	0.00000
chr1	153347960	rs779301061	G	GGA	-0.4162	0.2022	0.0396	35363	0.00036	0.00008
chr1	153351365	rs115697787	A	G	-0.0932	0.0448	0.0374	35362	0.00836	0.00000
chr1	153353113	rs745415623	T	C	1.1565	0.4320	0.0074	35363	0.00013	0.00000
chr1	153353210	rs780661487	G	C	0.1770	0.0903	0.0499	35363	0.00202	0.00000
chr1	153356544	rs59961408	A	G	-0.0255	0.0123	0.0381	35361	0.12511	0.20400
chr1	153356930	rs769941685	A	G	-0.2531	0.1283	0.0485	35360	0.00120	0.00000
chr1	153357702	rs12119788	A	G	0.0371	0.0132	0.0048	35360	0.10879	0.08500
chr1	153358301	rs112531265	A	G	-0.0878	0.0386	0.0228	35360	0.01117	0.00000
chr1	153358798	rs747411606	A	G	-0.3138	0.1345	0.0196	35360	0.00081	0.00000
chr1	153359373	rs2916193	A	G	-0.0477	0.0094	0.0000	35360	0.25783	0.34600
chr1	153359680	NA	CTT	CTT	0.0327	0.0090	0.0003	35350	0.35677	0.00000
chr1	153359680	rs1234133033	C	CTT	-0.1671	0.0845	0.0481	35350	0.00227	0.00000
chr1	153359680	rs1168990096, rs34039196	CT	CTT	-0.0471	0.0097	0.0000	35350	0.25293	0.00000
chr1	153361148	rs537228847	T	C	-0.2199	0.1002	0.0282	35360	0.00156	0.00000
chr1	153361324	rs2070864	G	A	-0.0456	0.0092	0.0000	35360	0.27329	0.02000
chr1	153361732	rs952053618	A	G	-0.4306	0.2077	0.0382	35360	0.00033	0.00000
chr1	153362049	rs1560833	A	G	-0.0466	0.0093	0.0000	35360	0.26674	0.42500
chr1	153362908	rs1560832	A	G	-0.0466	0.0093	0.0000	35360	0.26668	0.39200
chr1	153363542	rs724781	G	C	-0.0468	0.0093	0.0000	35360	0.26681	0.34800
chr1	153364102	NA	CAAAAAAA AAAA	CAA AAA AAA AAA CAA	0.0212	0.0107	0.0478	35355	0.18718	0.00000
chr1	153364102	rs1478401322, rs767275639	CAAAAAAA A	AAA AAA AAA AAA CAA AAA	-0.0527	0.0106	0.0000	35355	0.18232	0.00000
chr1	153364102	rs397863495, rs398049600,	CAAAAAAA AAA	CAA AAA AAA	0.0185	0.0084	0.0273	35355	0.46769	0.00000

		rs5777859		AAA AAA						
chr1	153364645	rs74807144	G	A	-0.1155	0.0266	0.0000	35360	0.02353	0.00000
chr1	153365467	rs3014874	A	G	-0.0537	0.0094	0.0000	35360	0.25791	0.30200
chr1	153365872	rs191112671	A	G	-0.1639	0.0696	0.0185	35360	0.00372	0.00000
chr1	153366275	rs3014875	A	G	-0.0282	0.0086	0.0010	35360	0.34949	0.11700
chr1	153366824	rs540566308	C	CAA G	-0.1351	0.0552	0.0144	35361	0.00569	0.00000
chr1	153370099	rs58644524	T	C	0.0273	0.0133	0.0397	35361	0.10771	0.09400
chr1	153370702	rs3014878	T	C	0.0275	0.0133	0.0384	35361	0.10772	0.07900
chr1	153371241	rs35195593	G	GC	0.0276	0.0133	0.0375	35361	0.10784	0.01000
chr1	153371405	rs3014879	G	A	0.0262	0.0132	0.0473	35361	0.10922	0.09800
chr1	153371466	rs12033317	A	G	0.0276	0.0133	0.0375	35361	0.10784	0.09600
chr1	153371894	NA	CATGC	CAT GC	-0.0277	0.0133	0.0372	35361	0.10763	0.00000
chr1	153371894	rs3014880	CATGG	CAT GC	0.0277	0.0133	0.0372	35361	0.10763	
chr1	153372160	rs3006475	C	A	0.0274	0.0133	0.0388	35362	0.10780	0.08050
chr1	153372439	rs3014881	G	A	0.0260	0.0132	0.0488	35362	0.10916	0.10500
chr1	153373217	NA	ATATATAT A	ATAT ATAT A	0.1288	0.0424	0.0024	35362	0.00921	0.00000
chr1	153373217	rs542546103	ATATATA	ATAT ATAT A	-0.1544	0.0476	0.0012	35362	0.00747	0.00000
chr1	153373435	rs2916191	C	T	0.0260	0.0132	0.0486	35362	0.10917	0.08200
chr1	153373787	rs4772	G	A	0.0260	0.0132	0.0485	35362	0.10917	0.07100
chr1	153376634	rs3006476	A	C	0.0272	0.0133	0.0409	35362	0.10751	0.02900
chr1	153377394	rs577781525	ATCCTGAG ATGTT	ATCC TGA GAT GT	0.1063	0.0460	0.0208	35360	0.00877	0.00000
chr1	153377731	rs141832834	A	AG	0.0271	0.0133	0.0419	35362	0.10740	0.01000
chr1	153379651	rs3014885	T	C	0.0272	0.0133	0.0409	35362	0.10754	0.10500
chr1	153379784	NA	ACACCTAG GGTGGCGG CGGCTCCTT GGCAG	ACA CCTA GGG TGG CGG CGG CTCC TTGG CAG ACA CCTA GGG TGG CGG CGG CTCC TTGG CAG CGTC	-0.0259	0.0132	0.0498	35362	0.10892	0.00000
chr1	153379784	rs374815343, rs555202163	A	CGG CGG CTCC TTGG CAG CGTC	0.0259	0.0132	0.0498	35362	0.10892	0.00000
chr1	153380435	NA	CGTCTCGG	TCG G CGTC	-0.0279	0.0132	0.0346	35362	0.10976	0.00000
chr1	153380435	rs377316404	CGTCTCGG A	CGTC TCG G	0.0279	0.0132	0.0347	35362	0.10975	0.00000

The 20 SNPs with MAF > 0.01 in the East Asian population are in bold.

Supplemental Figure 16 The association between S100A8/A9 genetic score and S100A8/A9 plasma concentrations in the validation cohort

Minor:

Lines 95-66, line 135: what are “genetic instruments”?

Reply, Genetic instruments are cis-pQTLs or genetic risk score associated with S100A8/A9 level.

Line 106: AMI patients are called ACS patients in Fig. S1. What is it?

Reply, This is a mistake, and we have corrected it in the revised version.

Line 100: what are “early-onset AMI patients”?

Reply, In validation cohort, the age of AMI patients is < 45 years old. Therefore, we call them early-onset AMI patients. According to previous studies, premature AMI was defined as the occurrence of an acute myocardial infarction before age 45.⁹

Lines 238-240: Supplemental Figures 4, 5 do not show unsupervised clustering data.

Reply, We have corrected this mistake. “Uniform manifold approximation and projection clustering based on single-cell RNA sequencing of peripheral blood cells (PBC) from patients with AMI with or without HF and HCs (n=2) identified 11 main cell subtypes.” (*Revised position: Page 6, Lines 16~19 in Result section*)

Line 239: n=2 contradicts Fig. S2 (3 x n=10).

Reply, We have corrected this mistake in the figure (revised Supplementary Figure 1).

Lines 243-244: „disease course“(implying serial measurements?) is not studied.

Reply, We have deleted the “disease course”. “Patients with AMI had reduced activated and memory CD4+ T cells and increased neutrophils and effector CD8+/cytotoxic T cells compared with HCs.” (*Revised position: page 6, line 20~22 in Result section*)

Lines 252-253: While the basic science literature indicates that neutrophils are an important source of S100A8/A9, the authors cannot conclude that this is the case also in their patients (they only studied RNA and only inflammatory cells, leaving the possibility that other cells types contributed as well). Line 330 also needs to be rewritten accordingly.

Reply, We have rewritten this part. We deleted “implying that neutrophils are the primary source of serum proteins for HF” in original Lines 252-253 and revised Line 330 into “particularly S100A8/A9 expression in neutrophils, was associated with post-AMI HF development”. (*Revised position: Page 13, Lines 3-4 in Discussion section*).

Lines 253-254: it is not true that S100A8/A9/A12 was exclusively produced by neutrophils (Figure S7, Fig. 2F).

Reply, We have changed this description into “S100A8/A9/A12 was mainly produced by neutrophils among the PBC” (*Revised position: Page 7, Line 6 in Result section*).

Line 265: hazard ratio for what?

Reply, hazard ratio for post-AMI HF. We revised into “Hazard ratio (HR) for post-AMI HF per standard deviation (SD) of S100A8/A9” (*Revised position: Page 7, Lines 18-19 in Result section*).

Lines 271-273: S100A12 was not associated with the combined endpoint also in univariate analysis.

Reply, We have changed this description. “HF-S100A12 association was still not significant after adjustment (HR per SD for model 3:1.08 [95% CI:0.95–1.22], P=0.261).” (*Revised position: page 7, Lines 24-25 and Page 8, line 1 in Result section*)

Lines 340-341: „the prognostic role of S100A8/A9 in AMI remained unestablished“. Not true, Marinkovic et al. (Ref. 15) have reported that plasma S100A8/A9 during the acute post-ischemic phase is correlated with hospitalization for heart failure and cardiac function at 1 year after ACS. This limits the ‘novelty’ of their biomarker observations.

Reply, Marinkovic et al. as well as our have reported that elevated S100A8/A9 is correlated with HF events in ischemia patients^{2, 3}, but these studies had certain limitations, including its small sample size, the lack of validation, and no significant result after adjustment for multiple factors. Therefore, the prognostic value of S100A8/A9 in AMI needs to be further established. (*Revised position: Page 13, Lines 9-15 in Discussion section*).

Line 350: what is The triangulation of evidence?

Reply, The triangulation of evidence means exposure factor, genetic instrument, and outcome in MR. We have revised this sentence into “MR analyses of S100A8/A9 enhanced the significance of our compared to other observational studies on prognostic markers.” (*Revised position: Page 14, Lines 1-2 in Discussion section*).

Lines 421-422: why young AMI patients?

Reply, In older patients, the prognosis of AMI is influenced by many other diseases, and the role of genetics in S100A8/A9 levels is also influenced by more confounding factors. Therefore, we

selected patients with early-onset AMI in the validation cohort to reduce confounding influences and highlight the role of genetic variants. (*Revised position: Page14, Lines 24-25 and Page15, Lines 1-2 in Discussion section*).

|

Reviewer #4

Summary: The authors perform a study aiming to identify proteins casual of increased risk of heart failure (HF) in patients with acute myocardial infarction (AMI). To do this, they identify candidate proteins for follow-up in a small cross-section cohort of 30 people: 10 AMI patients who developed HF during their hospital stay, 10 AMI patients who remained HF free at hospital discharge, and 10 healthy controls. After performing analyses of differential protein expression for ~1,000 proteins, they narrow down to two candidate targets; SA100-A8/A9 and S100A12. In a much larger prospective study of 1,062 AMI patients, they found that only SA100-A8/A9 and not S100-A12 measured at admission were predictive of increased risk of heart failure over a median of 4.2 years of follow-up. They further show this association replicated in an independent cohort of 1,183 patients with early onset AMI, and perform mediation analysis using a genetic score constructed from cis-pQTLs for S100-A9 identified in the deCODE pQTL mapping study, finding evidence suggesting that SA100-A8/A9 is a causal risk factor of heart failure development in AMI patients.

Reply, We greatly appreciate the reviewer's constructive suggestions that aimed to improve the quality of our study. We have specifically addressed each concern below.

Major comments:

I have several comments related to the causal inference:

1. First, I would argue that using Mendelian Randomization to describe the analysis performed could be misleading, and instead should be described instead as mediation analysis. This is an alternative framework for inferring causality that is quite different to most Mendelian Randomization methods readers would be more familiar with.

Reply 1, We apologize for our inaccurate previous description concerning Mendelian Randomization analysis.

We performed one-sample Mendelian Randomization analysis in validation cohort. S100A8/A9 genetic score were calculated for each individual in validation cohort. For a 1-SD increase in S100A8/A9 genetic scorers, the incidence of S100A8/A9 levels exceeding 4,877 ng/mL (cutoff value for high risk of HF) increased by 43% both in raw data and after adjusting for sex and age. A genetically predicted S100A8/A9 values were associated with post-AMI HF (OR per-SD: 1.20 [95% CI: 1.003–1.44], P=0.047), after adjusting for age, sex, SBP, Killip classification at admission, fasting glucose, creatinine, left main artery disease, neutrophil count, cTnI, BNP, CRP, LVEF and estimated infarct size (CK-MB AUC₀₋₇₂). (*Revised position: Page 11, Lines 24~25 and Page 12, Lines 1~4 in Result section*). In order to more intuitively elucidate the potential causal relationship between genetic score, S100A8/A9 levels, and heart failure, we have drawn a schematic diagram as below.

2. I would strongly recommend the authors redo their mediation analysis with the medflex R

package instead of the mediation R package. A major pitfall of the mediation R package as applied to the described analysis here is that by default it will be performing a counterfactual analysis of people with value of 0 for their S100A8/A9 genetic score to people with value of 1 for their S100A8/A9 genetic score. In contrast, the medflex package will perform counterfactual analysis over the full distribution of the S100A8/A9 genetic score, and as an added benefit, will also provide the estimated causal effect on a biologically meaningful scale (e.g. log OR per SD increase)

Reply 2, As Reviewer suggested, we perform Mediation analysis using the medflex R package and found that the association between genetic score for S100A8/A9 and HF events was mediated through circulating S100A8/A9 levels. The total effect of genetic score for S100A8/A9 was OR 1.21 (95% CI 1.001-1.45) which decomposed into direct effect OR 1.09 (95% CI 0.92-1.30) and indirect effect OR 1.10 (95% CI 1.03-1.17).

However, we seriously consider the Reviewer's comments and consult with two statistical experts (listed in the Acknowledgement). We realized that the present study aims to determine the causal relationship between S100A8/A9 levels and post-AMI HF using genetic score as an instrument, rather than to clarify if the genetic factors affect post-AMI HF through S100A8/A9. Therefore, mediation analysis is not suitable for this study and has been removed in the revised manuscript.

3. While cis-pQTLs provide a strong prior on the genetic score influencing HF risk only through S100A8/A9, false positive causal inferences are still possible e.g. due to horizontal pleiotropy or impacts of genetic variation on protein binding efficacy (see Zheng et al. Nature Genetics 2020). Evidence for a causal effect could be strengthened by adding colocalization analyses to assess whether the pQTL signal and SNP-to-HF association arise from one (or more) shared causal variant(s).

Reply 3, We strongly agree with the Review's opinion that colocalization analysis is necessary. Because we only genotyped six cis-pQTLs of S100A8/A9, colocalization analyses cannot be conducted in our cohort. We identified the SNPs associated with post-AMI HF in 1141 patients from UKB database, then assessed whether the cis-pQTLs of S100A8/A9 and SNPs-to-HF within 20 kb on either side of S100A9 gene arise from shared causal variants using Coloc R package. The results showed that PP.H0.abf = 6.56e-03, PP.H1.abf = 9.75e-01, PP.H2.abf = 3.35e-05, PP.H3.abf = 4.97e-03, and PP.H4.abf = 1.33e-02. Because of the small sample size used in association studies of SNPs and post-AMI HF, the power of this analysis was limited. (*Revised position: Page 12, Lines 15~19 in Result section*).

4. It would also be useful to see whether the causal effect of S100A8/A9 on risk of heart failure is significant using the more conventional two-sample Mendelian Randomization analysis, i.e. by overlaying the selected cis-pQTLs with GWAS summary statistics for Heart Failure. (E.g. Levin et al. Nature Communications 2022 have published GWAS summary statistics for Heart Failure in the GWAS Catalog in a cohort included ~260,000 participants of East Asian ancestries). The R package TwoSampleMR could be used to do this analysis. Following comment 4 above, it would be also interesting to see the results of two-sample Mendelian Randomization using a large Heart Failure GWAS to see whether the causal effect observed in patients with acute MI (assuming its robust) extends more generally to heart failure more

broadly.

Reply 4, We appreciated the Reviewer’s excellent suggestion. we conducted a two-sample MR analysis and verified the causal effect of S100A8/A9 on post-AMI HF (**Supplemental Figure 17**). In 1114 AMI patients from UKB database, due to the absence of rs12119788, S100A8/A9 genetic scores were composed other 5 SNPs. A genetically instrumented per-SD S100A8/A9 was associated with higher odds of post-AMI HF in Wald ratio analysis (OR: 2.06 [95% CI: 1.25–3.39]; P = 0.004) and inverse variance-weighted (IVW) analysis (OR: 1.55 [95% CI: 1.15–2.09]; P = 0.004). The Egger method provided no evidence of pleiotropy (Egger’s intercept, P = 0.653). We further cross-validated this finding in general HF using data from the finn-b-I9_HEARTFAIL study (Supplemental Figure 17). Using cis-pQTLs effect estimates on S100A8/A9 levels from the validation cohort of 1,043 patients with AMI, increased genetically predicted S100A8/A9 levels were associated with increased risk of HF (IVW estimate of OR per-SD:1.04 [95% CI:1.01–1.08]; P = 0.016) (*Revised Table 3*). (*Revised position: Page 12, Lines 5~14 and Lines 20~24 in Result section*).

Corresponding study design, data collection and two-sample MR analysis were provided in Supplemental Material (*Page 3, Lines 13-20, Page 4, Lines 15-22, and Page 14, Lines 1~21*).

Supplemental Figure 17. Diagram of two-sample Mendelian randomization framework in the present study.

Revised Table 3. Association of genetically determined plasma S100A8/A9 with risk of HF events in one-sample and two-sample MR

Methods	Genetic tools	Odds ratio (95%CI)	P-value
One-sample MR			
2SLS	S100A8/A9 genetic score	1.20 (1.003-1.44)	0.047
Two-sample MR			
Post-AMI HF			
Wald ratio	S100A8/A9 genetic score	2.06 (1.25-3.39)	0.004
IVW	cis-pQTLs for S100A8/A9	1.55 (1.15-2.09)	0.004

(rs59961408, rs1560832, rs3014874,
rs3014875, rs12033317)

MR-Egger regression (intercept = -0.10, P value = 0.744)

General HF

cis-pQTLs for S100A8/A9
IVW (rs59961408, rs12119788, rs1560832, 1.04 (1.01-1.08) 0.016
rs3014874, rs3014875, rs12033317)

MR-Egger regression (intercept = 0.001, P value = 0.894)

CI, confidence interval; HF, heart failure; IV, instrumental variable; IVW, inverse-variance weighted; MR, Mendelian randomization; OR, odds ratio; SNPs, single nucleotide polymorphisms; UKB, UK Biobank; 2SLS, Two Stages Least Square.

5. Regarding the identity of the target S100A8/A9, can the authors clarify whether this measurement is a protein complex of the A8 and A9 proteins? Or is it the case that the protein assays cannot easily distinguish between the two proteins?

Reply 5, S100A8 and S100A9 exist in several forms but preferentially form heterodimeric complex of S100A8/A9, which is the most abundant and stable form and necessary for their biological effects. Therefore, we detect the heterodimeric complex of S100A8/A9 in discovery and validation cohort using ELISA kit. The assay employed a quantitative sandwich enzyme immunoassay technique. Briefly, a monoclonal antibody specific to the human S100A8/S100A9 heterodimer was pre-coated onto a microplate. Subsequently, the samples were pipetted into the wells. After washing away the unbound substances, an enzyme-linked monoclonal antibody specific for the human S100A8/S100A9 heterodimer was added to each well. All steps strictly followed the operation procedure of ELISA protocol (**Revised position: Page 8, Lines 21~25 and Page 9, Lines 1~2 in Supplementary Material**).

6. Further to that comment, I note that the deCODE proteomics platform also includes a measurement for S100A8/A9 as well as S100A9 alone, but that the S100A8/A9 measurement did not have any cis (or trans) pQTLs.

Reply 6, To construct a genetic instrument for S100A8/A9, we search for GWAS summary statistics of plasma proteins for all 4,907 aptamers at <https://www.decode.com/summarydata/>. There are five files related to S100A8 and S100A9, including: “5339_49_S100A9_calgranulin_B.txt.gz”, “Proteomics_PC0_17145_1_S100A8_S100A9_S100A8_S100A9.txt.gz”, “Proteomics_PC0_5339_49_S100A9_calgranulin_B.txt.gz”, “Proteomics_SMP_PC0_17145_1_S100A8_S100A9_S100A8_S100A9.txt.gz”, and “Proteomics_SMP_PC0_5339_49_S100A9_calgranulin_B.txt.gz”. However, only the GWAS summary statistics for S100A9 “5339_49_S100A9_calgranulin_B.txt.gz” can be downloaded. We did not find the GWAS summary statistics for S100A8/A9 or S100A8.

We choose pQTLs of S100A9 on behalf of genetic instrument of S100A8/A9 based on the following 4 reasons: i) S100A9 regulates S100A8/A9 complex functions through various mechanisms like protecting S100A8 from degradation. ii) We observed that S100A9 levels was strongly correlated with S100A8/A9 levels in plasma ($r = 0.92$, $P < 0.01$). iii) The genotype-tissue expression (GTEx) portal strongly supported that 17/20 pQTLs of S100A9 were associated with

differential S100A8/A9 mRNA expression; iv) In a general Chinese population (n=588), the increase in the S100A8/A9 genetic score composed with 6 pQTLs of S100A9 was significantly associated with the high-risk S100A8/A9 levels (OR per SD: 1.40 [95% CI: 1.11–1.76], P = 0.004).

7. I do not have the relevant expertise to critique the single-cell transcriptomics analysis, but I'm somewhat concerned by the very small sample size here (N=6). It's also not clear to me that these experiments meaningfully add to the conclusions of the paper, so I would suggest cutting these analyses.

Reply 7, Considering that the single-cell transcriptomics analysis provides detailed information, such as the change of leukocyte composition in post-AMI HF, the cellular localization of candidate inflammatory proteins, we have moved this part into the Supplementary Materials.

8. I think it's also worth noting in the discussion that even for the initial differential protein expression scan the small sample size was quite small (N=30), so there may be other causal proteins that are being missed due to low detection power in the initial screening step.

Reply 8, We completely agree the Reviewer's comments. Because protein profiling experiments are costly, we performed the initial screening step in a small number of subjects. We acknowledge that other causal proteins might be missed in underpowered studies. We have added this information into limitation (*Revised position: Page 17, Lines 9-12 in Discussion section*).

9. It's somewhat unfortunate that so much of the space taken by the Main Figures are dedicated to analyses in the very small cross-sectional cohort, when the most impactful results come from the rest of the analyses and are shown mostly in the Supplemental Figures. In particular, I would suggest including a main Figure that shows (1) the significant differences in S100A8/A9 levels between HF and non-HF cases in both the discovery and validation cohorts, (2) the relationship between the genetic score for S100A8/A9 and measured S100A8/A9 in the healthy controls. E.g. A figure combining the current Figure 3 with Supplemental Figures S8, S11-16, additionally showing the validity of the genetic score for S100A8/A9.

Reply 9, We appreciated the Reviewer's constructive suggestion. We reformatted Figures 1-3 in revised main text. Revised Figure 1 shows the study design. Revised Figure 2 shows the predictive value of S100A8/A9 for HF in discovery and validation cohort. Revised Figure 3 shows Incidence of HF events according to S100A8/A9 levels. Revised Table 3 shows the causal effect of S100A8/A9 on post-AMI HF.

10. Another point that needs expanding on in the discussion is the differences between the validation cohort and the discovery cohort. The validation cohort participants are unusually young for acute MI patients. Is it possible that these patients have very different characteristics in terms of heart failure risk to patients whose age of onset of acute MI are more typical, and in turn, possible that the observed causal effect of S100A8/A9 on HF risk only applies to people with early-onset MI?

Reply 10, Due to the fact that in older patients, the prognosis of AMI is influenced by many other elderly diseases, and the role of genetics in S100A8/A9 levels is also influenced by more confounding factors, we selected early-onset AMI patients in validation cohort to reduce

confounders' influence and highlight genetic variants' role.

As Reviewer suggested, we analyzed the differences between the discovery and validation cohort. Compared to the discovery cohort, the validation cohort had more male patients, higher systolic blood pressure, and fewer patients with a history of comorbidities. Several laboratory indicators such as glucose, lipids, creatinine, BNP, cTnI, and CRP in validation cohort were also significantly lower than that in discovery cohort (*Supplemental Table 4*). These differences are consistent with the previous report.¹⁰ (*Revised position: Page 5, Lines 11~17 in Result section*)

The validation cohort comprised unusually young patients with AMI. To exclude the possibility that the observed causal effect of S100A8/A9 on HF risk only applies to patients with early onset MI, we used the UKB cohort for external validation of the causal effect, in which the median age at AMI onset was 56.0 [52.6-59.2] in patients with HF and 55.7 [51.6-58.4] in patients without HF (*Supplemental Table 17*). There is a causal effect of S100A8/A9 on post-AMI HF in the UKB cohort, indicating that the observed causal effect of S100A8/A9 on HF risk is not limited to individuals with early-onset MI (*Revised position: Page 15, Lines 2~9 in Discussion section*).

Supplemental Table 4. Comparison of baseline information between the discovery cohort and the validation cohort

Variables	Discovery (n=1062)	cohort	Validation (n=1043)	cohort	P-value
Demographics					
Age (years)	59.0 (51.0-67.0)		40.0 (36.0-42.0)		<0.001
Male sex	851 (80.1)		951 (91.2)		<0.001
SBP (mm Hg)	120.0 (109.0-135.0)		125.0 (117.0-133.0)		<0.001
DBP (mm Hg)	73.0 (68.0-82.0)		79.0 (70.0-84.0)		<0.001
Current smoking	652 (61.4)		680 (65.2)		0.070
Killip classification					
I	879 (82.8)		971 (93.1)		<0.001
II	163 (15.3)		58 (5.6)		<0.001
III	20 (1.9)		14 (1.3)		0.325
Medical history					
Hypertension	621 (58.5)		513 (49.2)		<0.001
Hyperlipidemia	695 (65.4)		482 (46.2)		<0.001
Diabetes mellitus	355 (33.4)		213 (20.4)		<0.001
CAD	258 (24.3)		209 (20.0)		0.019
Biochemical					
Neutrophil counts ($\times 10^9/L$)	5.8 (4.6-7.9)		6.3 (4.7-8.3)		0.002
HDL cholesterol (mmol/L)	1.0 (0.9-1.2)		0.9 (0.8-1.1)		<0.001
LDL cholesterol (mmol/L)	2.8 (2.3-3.5)		2.3 (1.7-3.0)		<0.001
Fasting glucose (mmol/L)	6.7 (5.6-9.3)		5.5 (5.0-6.6)		<0.001
Creatinine ($\mu\text{mol/L}$)	73.3 (62.8-84.5)		71.3 (63.8-79.5)		0.004
Biomarkers					
cTnI (ng/mL)	1.5 (0.2-9.6)		1.2 (0.2-6.6)		0.095
BNP (pg/mL)	107.7 (42.0-243.2)		82.0 (28.0-189.4)		<0.001
hs-CRP (mg/L)	6.0 (2.1-19.1)		4.7 (1.7-13.5)		<0.001
S100A8/A9 (ng/ml)	3640.8 5068.8)	(2345.7-	3657.4 (1905.0-4877.0)		0.060
Overall lesion profiles					
Left main artery disease	32 (3.0)		40 (3.8)		0.300
2-vessel disease	311 (29.3)		192 (18.4)		<0.001
3-vessel disease	137 (12.9)		226 (21.7)		<0.001
Echocardiography					
Admission LVEF (%)	54.0 (48.0-58.0)		55.0 (50.0-60.0)		<0.001
Infarct size					
CK-MB-based estimation (ng*h/ml)	5022.3 7088.9)	(2014.6-	4456.1 (2319.1-6275.4)		0.002
Medication at discharge					
Aspirin	1038 (97.7)		1010 (96.8)		<0.001
P2Y12 receptor Inhibitor	1062 (100.0)		964 (92.4)		<0.001

Statin	1019 (96.0)	990 (94.9)	0.002
ACEI or ARB	606 (57.1)	449 (43.0)	<0.001
Beta-blockers	783 (73.7)	729 (69.9)	0.010
MRA	175 (16.5)	70 (6.7)	<0.001

Supplemental Table 17. Clinical characteristics in AMI patients from UKB

Variable	UKB (n = 1144)		P-value
	HF events (n = 224)	No-HF events (n = 920)	
Age (years)	56.0 (52.6–59.2)	55.7 (51.6–58.4)	0.062
Male sex (%)	157 (70.1)	649 (70.5)	0.894

11. The follow-up time for both the discovery and validation cohorts extends beyond 2020, presenting the possibility of confounding from the COVID19 pandemic on heart failure risk (e.g. due to differences in treatment quality/availability during periods of large amounts of COVID19 cases, differences in lifestyle/behaviour due to lockdowns, or differences arising due to SARS-CoV2 infection or exposure). It would be great to see a sensitivity analysis of the S100A8/A9 associations with heart failure risk when truncating the follow-up time to the start of 2020.

Reply 11, As suggested by Reviewer, we performed a sensitivity analysis of the S100A8/A9 associations with HF risk when truncating the follow-up time to the start of 2020. The results showed that the association between S100A8/A9 levels and HF risk was not affected by the COVID-19 pandemic (*Supplemental Table 12*). We added this result to the revised manuscript. (*Revised position: Page 9, Lines 23~25 and Page 10, Lines 1~3 in Result section*).

Supplemental Table 12. The association of S100A8/A9 and HF risk was not affected by the COVID19 pandemic

	HR (95%CI)	P-value
Discovery cohort (n=1062)		
S100A8/A9	1.70 (1.48-1.95)	<0.001
Adjusted for model 1	1.82 (1.58-2.10)	<0.001
Adjusted for model 2	1.90 (1.65-2.20)	<0.001
Adjusted for model 3	1.76 (1.52-2.05)	<0.001
Validation cohort (n=1043)		
S100A8/A9	2.81 (2.31-3.40)	<0.001
Adjusted for model 1	2.80 (2.31-3.39)	<0.001
Adjusted for model 2	2.54 (2.09-3.09)	<0.001
Adjusted for model 3	2.18 (1.78-2.67)	<0.001

Unadjusted and adjusted HRs from Cox proportional hazards regression analysis were shown for discovery-cohort (n=1062) and validation-cohort (n = 1043) patients with HF. S100A8/A9 was measured on admission. Model 1: adjusted for age and sex; model 2: adjusted for model 1+systolic blood pressure, Killip classification at admission, fasting glucose, creatinine, left main artery disease; model 3: adjusted for model 2+neutrophil count, cardiac troponin I, B-type natriuretic peptide, c-reactive protein, left ventricular ejection fraction at admission and estimated infarct size (CK-MB AUC₀₋₇₂)

Minor comments:

1. Can the authors clarify whether the study termination time (May 31st 2021) was the maximum follow-up for both the discovery and validation cohorts?

Reply, Yes, the follow-up ended at death or termination time (May 31st 2021), which is the maximum follow-up for both the discovery and validation cohorts.

2. Supplementary Figure 8 and Supplementary Figure 13 would be clearer if the median and interquartile range were shown above the scatterplot points.

Reply, We have added the median and interquartile range on the scatterplot points.

3. Supplementary Figure 2 does not match the methods and results text: Supplementary Figure 2 shows 30 participants had scRNA-seq data, whereas the text states only 6 participants had these data.

Reply, We have corrected this mistake in the Figure (Supplemental Figure 2).

REFERENCES

- Giudice V, Wu Z, Kajigaya S, Fernandez Ibanez MDP, Rios O, Cheung F, Ito S and Young NS. Circulating S100A8 and S100A9 protein levels in plasma of patients with acquired aplastic anemia and myelodysplastic syndromes. *Cytokine*. 2019;113:462-465.
- Marinković G, Grauen Larsen H, Yndigejn T, Szabo IA, Mares RG, de Camp L, Weiland M, Tomas L, Goncalves I, Nilsson J, Jovinge S and Schiopu A. Inhibition of pro-inflammatory myeloid cell responses by short-term S100A9 blockade improves cardiac function after myocardial infarction. *European heart journal*. 2019;40:2713-2723.

3. Li Y, Chen B, Yang X, Zhang C, Jiao Y, Li P, Liu Y, Li Z, Qiao B, Bond Lau W, Ma XL and Du J. S100a8/a9 Signaling Causes Mitochondrial Dysfunction and Cardiomyocyte Death in Response to Ischemic/Reperfusion Injury. *Circulation*. 2019;140:751-764.
4. McDonagh TA, Metra M, Adamo M, Gardner RS, Baumbach A, Böhm M, Burri H, Butler J, Čelutkienė J, Chioncel O, Cleland JGF, Coats AJS, Crespo-Leiro MG, Farmakis D, Gilard M, Heymans S, Hoes AW, Jaarsma T, Jankowska EA, Lainscak M, Lam CSP, Lyon AR, McMurray JJV, Mebazaa A, Mindham R, Muneretto C, Francesco Piepoli M, Price S, Rosano GMC, Ruschitzka F and Kathrine Skibelund A. 2021 ESC Guidelines for the diagnosis and treatment of acute and chronic heart failure. *European heart journal*. 2021;42:3599-3726.
5. Leukert N, Vogl T, Strupat K, Reichelt R, Sorg C and Roth J. Calcium-dependent tetramer formation of S100A8 and S100A9 is essential for biological activity. *Journal of molecular biology*. 2006;359:961-72.
6. Nagareddy PR, Sreejit G, Abo-Aly M, Jaggars RM, Chelvarajan L, Johnson J, Pernes G, Athmanathan B, Abdel-Latif A and Murphy AJ. NETosis Is Required for S100A8/A9-Induced Granulopoiesis After Myocardial Infarction. *Arteriosclerosis, thrombosis, and vascular biology*. 2020;40:2805-2807.
7. Ternant D, Ivanes F, Prunier F, Mewton N, Bejan-Angoulvant T, Paintaud G, Ovize M and Angoulvant D. Revisiting myocardial necrosis biomarkers: assessment of the effect of conditioning therapies on infarct size by kinetic modelling. *Scientific reports*. 2017;7:10709.
8. Levitov A, Frankel HL, Blaivas M, Kirkpatrick AW, Su E, Evans D, Summerfield DT, Slonim A, Breikreutz R, Price S, McLaughlin M, Marik PE and Elbarbary M. Guidelines for the Appropriate Use of Bedside General and Cardiac Ultrasonography in the Evaluation of Critically Ill Patients-Part II: Cardiac Ultrasonography. *Critical care medicine*. 2016;44:1206-27.
9. Collet JP, Zeitouni M, Procopi N, Hulot JS, Silvain J, Kerneis M, Thomas D, Lattuca B, Barthelemy O, Lavie-Badie Y, Esteve JB, Payot L, Brugier D, Lopes I, Diallo A, Vicaut E and Montalescot G. Long-Term Evolution of Premature Coronary Artery Disease. *Journal of the American College of Cardiology*. 2019;74:1868-1878.
10. Liu Q, Shi RJ, Zhang YM, Cheng YH, Yang BS, Zhang YK, Huang BT and Chen M. Risk factors, clinical features, and outcomes of premature acute myocardial infarction. *Frontiers in cardiovascular medicine*. 2022;9:1012095.

REVIEWER COMMENTS

Reviewer #1 (Remarks to the Author):

The authors addressed the reviewer's comments by supplementing some data experimentally and revising the manuscript carefully. The responses and the revised manuscript have been reviewed as satisfactory. It seems a good shape for publication in Nature Communication.

Reviewer #2 (Remarks to the Author):

This is the second time I am reviewing this article.

The authors have responded adequately to my questions.

I would have the following residual comments:

I would encourage the authors to highlight the lack of UACR data in the limitations and references both the article provided in the review and <https://jamanetwork.com/journals/jama/fullarticle/194038> which clearly shows the prognostic role of UACR in heart failure prognostication.

Reviewer #3 (Remarks to the Author):

Following my previous suggestion, the authors now focus on HF events (as meticulously defined in Table S3). Good, that patients have been so carefully followed-up. HF event numbers are presented in Table S2. Looking at these numbers, I find it amazing that non-HF-related deaths were rare and non-cardiac deaths exceedingly rare despite a median follow-up of 4.2 and 2.9 years in the discovery and validation cohorts, respectively. This contradicts published evidence from another contemporary STEMI patient cohort (Yamashita Y et al. on the CREDO-Kyoto AMI Registry Investigators. Cardiac and Noncardiac Causes of Long-Term Mortality in ST-Segment-Elevation Acute Myocardial Infarction Patients Who Underwent Primary Percutaneous Coronary Intervention. *Circ Cardiovasc Qual Outcomes*. 2017;10:e002790). Any idea why that is?

Line 32: HR per what?

Line 34: "reclassification of traditional risk factors" – wording?

Line 36: OR per what?

Line 69: biomarkers do not "enhance" the "diagnosis and prognosis"

Line 75: "genetically predicted portion of biomarkers" – wording?

Line 77, I suggest: ...identify biomarkers associated with HF development in patients...

Line 117, Table S5: I find it hard to believe that 0% of women but 92.9% of men are current smokers

Lines 124-5: "identify the biomarkers associated with HF in patients with AMI" – wording?

Page 6, bottom: considering that only 2 individuals per group have been studied, it is impossible to conclude that differences in cell numbers are real and not a play of chance.

I would like to learn more about the “AMI patients”. In Fig. S3 (and only there) are they identified as STEMI patients. Is that true? How were these patients treated? PCI? CABG? Culprit vessel?

Figure S6 is not called out in the text.

Line 154 (and Discussion) “expressed” (mRNA) not “produced”

Lines 164-5: “significantly greater” based on what statistical test?

Lines 169-70: “after adjustment” vs. “univariable analysis” seems to be a contradiction

Figure S12, Tables S4, S7, S13: how can AMI patients not have CAD?

Line 220: Figure S14 does not “support the predictive value of S100A8/A9”

Table S12: how do the data presented here rule out COVID19 influence?

Line 244: how were S100A9 levels measured?

Lines 301-302: “S100A8/A9 expression in neutrophils, was associated with post-AMI HF development” – this is not firmly established (n=2 in scRNA-seq study).

Line 317: “Circulating S100A8/A9 released by neutrophils.” Meaning what?

Line 387: I suggest “post-discharge...”

Lines 389-391: pure speculation. Should rephrase.

Line 398: required to elucidate...

The manuscript needs to be carefully checked by a native speaker.

Reviewer #4 (Remarks to the Author):

The authors have adequately responded to my original comments. In particular its great to see the overhauled study design with respect to causal inference and its encouraging to see the results replicate in two-sample Mendelian Randomization performed in post-AMI patients in UK Biobank. I only have a few remaining mostly minor comments.

With respect to my original major comments 1 and 2 regarding mediation analysis, I agree with the authors decision to replace this analysis entirely with Mendelian Randomization. Further, I wanted to note that it’s reassuring to see that the updated mediation analysis provided in the reviewer responses is consistent with new Mendelian Randomization analyses.

For Supplementary Figure 15, can the authors please clarify in the figure caption what is being shown in the heatmap in panel B? In particular it’s unclear to me what the numbers in each cell are meant to represent.

For the Mendelian Randomization analyses, it would be useful to also see dose-response curves for the results; i.e. scatterplots comparing the effect size for each cis-pQTL on S100A8/A9 levels (x-axis) to the log odds (or hazard ratio) for each cis-pQTL on post-MI (or general) HF (y-axis), with a line of best fit drawn (where the slope is the causal estimate of S100A8/A9 levels on post-MI/general HF). Likewise, it would be useful to see some kind of figure comparing the S100A8/A9 genetic score to HF (perhaps similar to Figure 3, but showing quartiles of genetic score instead of quartiles of measured S100A8/A9). If there is space, these could perhaps be combined into another main figure; it would be nice for the final figure to show the causal effects of S100A8/A9 on post-MI HF.

I have several related comments on the colocalization analysis:

(1) First and most importantly, the fact the PP.H1.abf (the probability of only 1 of the 2 traits having a significant association at that locus) is the only posterior probability close to 1 indicates, as the authors rightly note in the text, that there is limited power for the analysis. Guidelines for colocalization analysis recommend only testing for colocalization where $P < 1e-6$ for both traits in question, so the authors could justifiably omit the results and simply note that due to the small sample size of the post-MI cohort (and even smaller number of subsequent HF cases) there is insufficient power to test colocalization in a post-MI setting

(2) It would be useful instead to test colocalization between the cis-pQTL signals and GWAS signals for heart failure more generally where sample sizes are much larger. Since the Mendelian Randomization results also support a causal role for S100A8/A9 in heart failure more generally (although with weaker effect), colocalization in that setting will still be useful in providing additional evidence that the pQTL and HF signals are arising from a shared causal variant, and these observations would also carry over to post-MI HF, or if they do not share a causal variant, there may be some sort of cryptic pleiotropy involved that's not being ruled out by the MR-Egger test.

(3) The way the colocalization analysis results are currently presented is difficult to read. It relies on the reader knowing the meaning of the abbreviations PP.H1.abf etc. The numbers are also currently presented in scientific format, which is not helpful as, unlike p-values, posterior probabilities close to 1 (e.g. >0.9) are of interest. In general the two numbers of interest are PP.H3.abf (the posterior probability that both traits are associated, but with different causal variants) and PP.H4.abf (the posterior probability that both traits are associated and share a causal variant). If these are both close to 0, then either there's not enough power to test the colocalization (i.e. if one of PP.H1.abf or PP.H2.abf are >0.9), or there's no association (i.e. if PP.H0.abf is > 0.9 , which we know is not true). When presenting the results, it's also useful to see a visual comparison of the pQTL and GWAS signals at the locus, i.e. by showing LocusZoom plots of the pQTL and GWAS signals side-by-side as a supp figure.

Reviewer #2 (Remarks to the Author):

This is the second time I am reviewing this article. The authors have responded adequately to my questions. I would have the following residual comments:

I would encourage the authors to highlight the lack of UACR data in the limitations and references both the article provided in the review and <https://jamanetwork.com/journals/jama/fullarticle/194038> which clearly shows the prognostic role of UACR in heart failure prognostication.

Reply: We appreciate all comments from Reviewer to make our manuscript better! As Reviewer suggests, we have highlighted the lack of UACR data in the limitations as follow: Lastly, the urine albumin-creatinine ratio (UACR) is a prognostic marker of adverse HF outcomes in patients with ACS with type 2 diabetes^{1 2}. We have not conducted this analysis because of the missing of UACR in present study. However, in our discovery and validation cohorts, 67% and 80% patients have no history of type 2 diabetes, respectively. (*Revised position: page 18, lines7-11 in Discussion section*).

Reviewer #3 (Remarks to the Author):

1, Following my previous suggestion, the authors now focus on HF events (as meticulously defined in Table S3). Good, that patients have been so carefully followed-up. HF event numbers are presented in Table S2. Looking at these numbers, I find it amazing that non-HF-related deaths were rare and non-cardiac deaths exceedingly rare despite a median follow-up of 4.2 and 2.9 years in the discovery and validation cohorts, respectively. This contradicts published evidence from another contemporary STEMI patient cohort (Yamashita Y et al. an the CREDO-Kyoto AMI Registry Investigators. Cardiac and Noncardiac Causes of Long-Term Mortality in ST-Segment-Elevation Acute Myocardial Infarction Patients Who Underwent Primary Percutaneous Coronary Intervention. Circ Cardiovasc Qual Outcomes. 2017;10:e002790). Any idea why that is?

Reply 1: We greatly appreciate the Reviewer’s constructive suggestions. Firstly, we carefully analyze the cause of rare non-cardiac deaths. Because there are young participants (≤ 45 years old) and short follow-up time (2.9 years [IQR, 1.6–4.2]) in validation cohort, we emphasize to compare the data in discovery cohort with CREDO-Kyoto AMI cohort. There are three reasons for lower non-cardiac in our cohort:

- i) Different age of patients. The age of patients in discovery cohort was younger than that in CREDO-Kyoto AMI cohort (58.9 ± 11.7 vs. 67.6 ± 12.3 years). It is noteworthy that only 10% of patients in our cohort were over 75 years old, while the ratio was 31% in their cohort. The seventh China Census report notes that the annual all-cause mortality rate of people over 75 years old is nearly 10 times that of people aged 50-70.
- ii) Different recruitment and exclusion criteria. We excluded patients with cardiogenic shock (Killip class IV) on admission, active infection, systemic inflammatory disease, known malignant disease, as well as surgical procedure within the previous 3 month. These patients were recruited in Yamashita Y’s study, especially 8% patients have malignancy.
- iii) Different medical history. The proportions of patients with low eGFR (< 30 mL/min per m^2) or hemodialysis were lower in our cohort than that in CREDO-Kyoto AMI cohort (1% vs. 5%). The deaths due to malignancie, infection/sepsis and renal failure were 0% and 5.7% in two cohorts, respectively.

	Discovery cohort (n = 1062)	CREDO-Kyoto AMI cohort (n = 3942)
Non-cardiac deaths	3 (0.28%)	347 (8.8%)
Stroke	2 (0.18%)	28 (0.7%)
Malignancies	1 (0.1%)	116 (2.9%)
Infection/Sepsis	0	95 (2.4%)
Renal failure	0	13 (0.3%)
Respiratory insufficiency	0	22 (0.6%)
Others	0	73 (1.9%)
Baseline Characteristics		

Age (years)	58.9±11.7	67.6±12.3
Age ≥ 75 years	108 (10.2%)	1227 (31%)
eGFR <30 mL/min/m ² , no dialysis	10 (1%)	163 (4.1%)
Hemodialysis	0	55 (1.4%)
Malignancy	0	318 (8.1%)

Next, we explain why there are fewer patients with non-HF-related cardiac death. We think that the main reason for rare non-HF related deaths is different definition. Definition of HF-related deaths were more stringent in CREDO-Kyoto AMI cohort than that in our cohort. In Yamashita Y’s study, many types of cardiac deaths, such as cardiogenic shock, cardiopulmonary arrest (CPA) on arrival, ventricular arrhythmia et al, were classified as non-HF related death. We define HF-related cardiac death as death in the context of clinically worsening symptoms and/or signs of heart failure with no other apparent cause, death as a consequence of a surgical procedure to treat heart failure, or death after referral to hospice for heart failure. In our study, cardiogenic shock, CPA on arrival et al are defined as HF-related deaths. Thus, there are more HF-related deaths (7.8% vs. 1.4%) and less non-HF related deaths (1.7% vs. 11.1%) in our cohort compared with CREDO-Kyoto AMI cohort.

Causes of cardiac deaths	Discovery cohort (n = 1062)	CREDO-Kyoto AMI cohort (n = 3942)
Cardiac death	101 (9.5%)	493 (12.5%)
HF-related deaths	83 (7.8%)	56 (1.4%)
non-HF related deaths	18 (1.7%)	437 (11.1%)
Cardiogenic shock	12 (1.1%)	131 (3.3%)
CPA on arrival	\	28 (0.7%)

2, Line 32: HR per what?

Reply 2: We have revised this position to be “HR per SD”. (*Revised position: Page2 line9 in Abstract section*)

3, Line 34: “reclassification of traditional risk factors” – wording?

Reply 3: We have revised this position to be “The addition of the S100A8/A9 improved the risk estimation based on traditional risk factors”. (*Revised position: Page2 lines 11-12 in Abstract section*)

4, Line 36: OR per what?

Reply 4: We have revised this position to be “OR per SD”. (*Revised position: Page2 line14 in Abstract section*)

5, Line 69: biomarkers do not “enhance” the “diagnosis and prognosis”

Reply 5: We have revised this position to be “Proteomic profiles represent sources of new candidate biomarkers with diagnostic and prognostic value”. (*Revised position: Page3 lines 20-21 in Introduction section*)

6, Line 75: “genetically predicted portion of biomarkers” – wording?

Reply 6: We have revised this position to be “genetically predicted portion of biomarker”. (*Revised position: Page4 lines 1-2 in Introduction section*)

7, Line 77, I suggest: ...identify biomarkers associated with HF development in patients...

Reply 7: We have revised this position to be “In this study, we aimed to identify biomarkers associated with HF development in patients with AMI”. (*Revised position: Page4 lines 4-5 in Introduction section*)

8, Line 117, Table S5: I find it hard to believe that 0% of women but 92.9% of men are current smokers

Reply 8: We apologize for this mistaken description. In fact, the data include the current smokers and those who have any smoking history. The current smokers have been corrected as “Former and current smokers”. (*Revised position: Page22 line2 in Supplemental Material*)

9, Lines 124-5: “identify the biomarkers associated with HF in patients with AMI” – wording?

Reply 9: We revised this position to “in patients without HF vs. those with HF development”. (*Revised position: Page6 line1-2 in Result section*)

10, Page 6, bottom: considering that only 2 individuals per group have been studied, it is impossible to conclude that differences in cell numbers are real and not a play of chance.

Reply 10: We agree with Reviewer’s opinion. We revised this position to be “Patients with AMI have a declining trend in activated and memory CD4⁺ T cells and a rising trend in neutrophils and effector CD8⁺/cytotoxic T cells compared to HCs. Due to the limited number of samples, this observation needs to be verified in a further study”. (*Revised position: Page6 lines 22-25 in Result section*)

11, I would like to learn more about the “AMI patients”. In Fig. S3 (and only there) are they identified as STEMI patients. Is that true? How were these patients treated? PCI? CABG? Culprit vessel?

Reply 11: We apologized for this mistake and revised into “patients with AMI without HF (no-HF), and patients with AMI (HF)”. These patients underwent primary PCI. The information of culprit vessel has been added into *Supplemental Table 1*. (*Revised position: Page17 line3 in Supplemental Material*)

Supplemental Table 1 Clinicopathological characteristics of patients with AMI (n = 24) and healthy controls (HCs) (n = 12) in the first screening stage

Variable	HC (n=12)	AMI (n =24)		P-value
		No-HF events (n=12)	HF events (n=12)	
Culprit lesion profiles				/
LAD	/	6 (50.0)	6 (50.0)	1.000
LCX	/	2 (16.7)	2 (16.7)	1.000
RCA	/	4 (33.3)	4 (33.3)	1.000
PCI	/	12 (100.0)	12 (100.0)	/

LAD, left anterior descending artery; LCX, left circumflex artery; RCA, right coronary artery; PCI, percutaneous coronary intervention.

12, Figure S6 is not called out in the text.

Reply 12: We have called out Figure S6 in last version as follows: “Uniform manifold approximation and projection clustering based on single-cell RNA sequencing of peripheral blood cells (PBCs) from patients with AMI with or without HF and HCs (n=2) identified 11 main cell subtypes (*Supplemental Figure 5*) based on their marker genes (*Supplemental Figure 6*)” (*Revised position: Page6 line17-20 in Result section*)

13, Line 154 (and Discussion) “expressed” (mRNA) not “produced”

Reply 13: We have revised into “S100A8/A9/A12 was mainly expressed by neutrophils.” (*Revised position: page7, lines8-9 in Result section.*), “Although neutrophils express S100A8/A9/12” (*Revised position: page16, line19 in Result section.*)

14, Lines 164-5: “significantly greater” based on what?

Reply 14: Sorry for this inaccurate description. We have changed into “The levels of S100A8/A9 and S100A12 were increased by 1.5-fold and 1.1-fold in patients with HF compared to those without HF, respectively” (*Revised position: page7, lines19-20 in Result section*)

15, Lines 169-70: “after adjustment” vs. “univariable analysis” seems to be a contradiction

Reply 15: Sorry for this inaccurate description. We have changed into “In multivariate analysis, the HF-S100A8/A9 association remained significant after adjustment for sex and significant clinical variables chosen from the univariable analysis (P < 0.05)”. (*Revised position: page7, lines24-25 and page8, line1 in Result section*)

16, Figure S12, Tables S4, S7, S13: how can AMI patients not have CAD?

Reply 16: CAD in Figure S12, Tables S4, S7, S13 is the medical history of CAD, which refers to the CAD events before admission. As shown in the table below, the proportion

of patient with a history of CAD in AMI cohorts were 23%~33% in previous reports, which is similar to 30% and 20% in discovery and validation cohort.

Reference	Cohort	Patients with medical history of CAD
J Am Coll Cardiol. 2014 Oct 21;64(16):1698-707.	1,148 STEMI and non-STEMI patients	379 (33%)
J Am Heart Assoc. 2020 Nov 3; 9(21): e016623.	913 AMI patients in pregnancy	307 (33.6%)
J Am Coll Cardiol. 2019 Aug 13;74(6):774-782.	1146 ACS patients	269 (23.5%)
Discovery cohort in this study	1062 AMI patients	319 (30.0%)
Validation cohort in this study	1043 AMI patients	209 (20%)
Total	2105 AMI patients	528 (25.1%)

17, Line 220: Figure S14 does not “support the predictive value of S100A8/A9”

Reply 17: We observed that the S100A8/A9 concentrations were higher in Killip class II and III patients than in Killip class I patients in the combined cohort. (*Supplemental Figure 14*). Due to the limited sample size in patients with Killip class III, a rising trend was only observed between Killip class III and class II. (*Revised position: page9, lines24-25 and page10, lines1-2 in Result section*)

Supplemental Figure 14 S100A8/A9 plasma concentrations in each Killip class on admission in combined cohorts

S100A8/A9 levels were show in combined cohorts according to Killip classI, II and III. Green: Patients with Killip class I; blue: Patients with Killip class II; Red: Patients with Killip class III.

18, Table S12: how do the data presented here rule out COVID19 influence?

Reply 18: We agree with the Reviewer this may not be very accurate. The ideal way is to analyze the HF-S100A8/A9 associations in patients infected or non-infected COVID19. However, we have no access to the information for COVID19 infection in patients. We performed a sensitivity analysis of the HF-S100A8/A9 associations before and after COVID-19 pandemic (January 1, 2020). S100A8/A9 levels were still independently associated with HF risk (*Supplemental Table 12*). (*Revised position: page10, lines5-7 in Result section*)

Supplemental Table 12 The association of S100A8/A9 and heart failure risk before and after January 1st 2020

	Before COVID-19		After COVID-19	
	HR (95%CI)	P -value	HR (95%CI)	P -value
Discovery cohort (n=1062)				
S100A8/A9	1.71 (1.41-2.07)	<0.001	3.65 (2.54-5.24)	<0.001
Adjusted for model 1	1.79 (1.47-2.17)	<0.001	3.86 (2.66-5.60)	<0.001
Adjusted for model 2	1.83 (1.50-2.24)	<0.001	3.96 (2.70-5.81)	<0.001
Adjusted for model 3	1.76 (1.43-2.17)	<0.001	4.22 (2.81-6.31)	<0.001
Validation cohort (n=1043)				
S100A8/A9	2.91 (2.07-4.09)	<0.001	1.99 (1.28-3.11)	0.002
Adjusted for model 1	2.91 (2.06-4.10)	<0.001	1.96 (1.25-3.08)	0.003
Adjusted for model 2	2.80 (1.97-3.96)	<0.001	1.93 (1.23-3.03)	0.004
Adjusted for model 3	2.40 (1.68-3.42)	<0.001	1.95 (1.22-3.12)	0.005

Unadjusted and adjusted HRs from Cox proportional hazards regression analysis were shown for discovery-cohort (n=1062) and validation-cohort (n = 1043) patients with HF. S100A8/A9 was measured on admission. Model 1: adjusted for age and sex; model 2: adjusted for model 1+systolic blood pressure, Killip classification at admission, fasting glucose, creatinine, left main artery disease; model 3: adjusted for model 2+neutrophil count, cardiac troponin I, B-type natriuretic peptide, c-reactive protein, left ventricular ejection fraction at admission and estimated infarct size (CK-MB AUC₀₋₇₂).

19, Line 244: how were S100A9 levels measured?

Reply 19: S100A9 levels were measured using a standard ELISA kit (S100A9: RayBiotech, Norcross, GA, USA) (*Revised position: page9, lines 3-4 in Supplemental Material*).

20, Lines 301-302: “S100A8/A9 expression in neutrophils, was associated with post-AMI HF development” – this is not firmly established (n=2 in scRNA-seq study).

Reply 20: We have revised this to be “high inflammation status, particularly increased S100A8/A9 level, was observed in patients with post-AMI HF.” (*Revised position: page13, lines19-21 in Discussion section*)

21, Line 317: “Circulating S100A8/A9 released by neutrophils.” Meaning what?

Reply 21: We have revised this to be “scRNA-RNA data showed that S100A8/A9 was mainly expressed by neutrophils.” (*Revised position: page14, line12-13 in Discussion section*)

22, Line 387: I suggest “post-discharge...”

Reply 22: We have revised this to be “post-discharge follow-up”. (*Revised position: page17, line8-9 in Discussion section*)

23, Lines 389-391: pure speculation. Should rephrase.

Reply 23: We completely agree Reviewer’s opinion and revised this sentence as follows: “Elevated S100A8/A9 levels drive the post-AMI progression of the inflammatory response and mitochondrial dysfunction. Several anti-inflammatory agents or mitochondria-targeting peptides are promising drugs for cardiovascular disease^{3, 4}, and these treatments may be more effective for patients with high S100A8/A9 levels”. (*Revised position: page17, lines 9-13 in Discussion section*).

24, Line 398: required to elucidate...

Reply 24: We have revised this to “required to elucidate”. (*Revised position: page17, line20 in Discussion section*).

25, The manuscript needs to be carefully checked by a native speaker.

Reply 25: The manuscript has been polished by native English editors. The proof material of language embellishment was shown below.

editage

Editing Certificate

This document certifies that the manuscript listed below has been edited to ensure language and grammar accuracy and is error free in these aspects. The logical presentation of ideas and the structure of the paper were also checked during the editing process. The edit was performed by professional editors at Editage, a brand of Cactus Communications. The author's core research ideas were not altered in any way during the editing process. The quality of the edit has been guaranteed, with the assumption that our suggested changes have been accepted and the text has not been further altered without the knowledge of our editors.

MANUSCRIPT TITLE

S100A8/A9 as a Prognostic Biomarker with Causal Effects for Post-acute Myocardial Infarction Heart Failure

AUTHORS

Yulin Li

ISSUED ON

November 25, 2023

JOB CODE

LRBZO_1_2

Prabh Grewal
Senior Vice President - Editage

editage | helping you
get published

Since 2002, Editage has helped over 430,000 authors publish around 1.2 million research papers in scholarly journals across over 1000 disciplines through editorial, translation, transcription, and publication support services. Editage is a brand of Cactus Communications (cactusglobal.com), a science communication and technology company.

GLOBAL :
+1(833) 979-0061 | request@editage.com

CHINA :
400-120-3020 或 021-6020-9400 |
fabiao@editage.cn

CACTUS.

Reviewer #4 (Remarks to the Author):

The authors have adequately responded to my original comments. In particular its great to see the overhauled study design with respect to causal inference and its encouraging to see the results replicate in two-sample Mendelian Randomization performed in post-AMI patients in UK Biobank. I only have a few remaining mostly minor comments. With respect to my original major comments 1 and 2 regarding mediation analysis, I agree with the authors decision to replace this analysis entirely with Mendelian Randomization. Further, I wanted to note that it's reassuring to see that the updated mediation analysis provided in the reviewer responses is consistent with new Mendelian Randomization analyses.

1, For Supplementary Figure 15, can the authors please clarify in the figure caption what is being shown in the heatmap in panel B? In particular it's unclear to me what the numbers in each cell are meant to represent.

Reply1: Thank you for the Review's reminder. We clarify what is being shown in the heatmap of panel B in the figure legend as follows: "Panel B shows the twenty cis-pQTLs selecting within 20 kb of either side of S100A9 and their linkage disequilibrium relationship calculated by Haploview software using CHB data as reference. The number in each cell is R^2 of its corresponding two SNPs, which is a parameter indicating the degree of linkage disequilibrium between two SNPs. The larger R^2 , the stronger the degree of linkage disequilibrium. Red color indicates stronger correlation." (*Revised position: Page57 lines2-6 in Supplemental Material*)

2, For the Mendelian Randomization analyses, it would be useful to also see dose-response curves for the results; i.e. scatterplots comparing the effect size for each cis-pQTL on S100A8/A9 levels (x-axis) to the log odds (or hazard ratio) for each cis-pQTL on post-MI (or general) HF (y-axis), with a line of best drawn (where the slope is the causal estimate of S100A8/A9 levels on post-MI/general HF). Likewise, it would be useful to see some kind of figure comparing the S100A8/A9 genetic score to HF (perhaps similar to Figure 3, but showing quartiles of genetic score instead of quartiles of measured S100A8/A9). If there is space, these could perhaps be combined into another main figure; it would be nice for the final figure to show the causal effects of S100A8/A9 on post-MI HF.

Reply2: As Reviewer suggested, we provided dose-response curves showing the causal effect of S100A8/A9 levels on post-AMI/general HF, with a line of best drawn (*Supplemental Figure 19*). The slope is the causal estimate of S100A8/A9 levels on post-MI/general HF. (*Revised position: page13, lines 2-5 in Result section*).

We also provided a Kaplan–Meier curve comparing the S100A8/A9 genetic score to HF (**Figure 3C**). At the median follow-up time, the patients in the higher-risk categories stratified by the quartile of S100A8/A9 genetic score exhibited a higher risk of post-AMI HF events. The stratification was less significant due to the limited patients at the late follow-up time. (*Revised position: page12, lines 3-7 in Result section*).

Supplemental Figure 19. Dose-response curves showing the causal effect of S100A8/A9 levels on post-MI/general HF

The strength of the association between HF events and each S100A9 SNPs on the y-axis against the S100A8/A9 levels association for each SNP on the x-axis. Results for all SNPs are shown for HF events in (A) validation cohort (192 HF events, 851 No-HF events), (B) UK Biobank (224 HF events, 920 No-HF events) and, (C) finn-b-I9_HEARTFAIL study (13,087 patients with HF, 19,5091 controls). The slopes of line represent the causal effect of inverse-variance weighted.

Figure 3. Incidence of HF events according to S100A8/A9 levels and S100A8/A9 genetic score

Similarly, the quartile of the S100A8/A9 genetic score was used to classify patients in the validation cohort into four risk categories (C).

3, I have several related comments on the colocalization analysis:

(1) First and most importantly, the fact the PP.H1.abf (the probability of only 1 of the 2 traits having a significant association at that locus) is the only posterior probability close to 1 indicates, as the authors rightly note in the text, that there is limited power for the analysis. Guidelines for colocalization analysis recommend only testing for colocalization where $P < 1e-6$ for both traits in question, so the authors could justifiably omit the results and simply note that due to the small

sample size of the post-MI cohort (and even smaller number of subsequent HF cases) there is insufficient power to test colocalization in a post-MI setting

Reply 3.1: We appreciate Reviewer's constructive suggestion. We revised the content of the colocalization analysis as follows: Colocalization analyses can strengthen the evidence for a causal effect. However, guidelines for colocalization analysis recommend only testing for colocalization where $P < 10^{-6}$ for both traits in question. Due to the small sample size of the post-AMI cohort, there is insufficient power to test colocalization in our cohort (*Revised position: page13, lines 6-9 in Result section*).

(2) It would be useful instead to test colocalization between the cis-pQTL signals and GWAS signals for heart failure more generally where sample sizes are much larger. Since the Mendelian Randomization results also support a causal role for S100A8/A9 in heart failure more generally (although with weaker effect), colocalization in that setting will still be useful in providing additional evidence that the pQTL and HF signals are arising from a shared causal variant, and these observations would also carry over to post-MI HF, or if they do not share a causal variant, there may be some sort of cryptic pleiotropy involved that's not being ruled out by the MR-Egger test.

Reply 3.2: Although the sample size for general HF is much larger, the minimum P-value of the SNPs within the target region is 0.0068, which does not meet the criteria for conducting colocalization analysis (*Revised position: page13, line9-12 in Result section*).

(3) The way the colocalization analysis results are currently presented is difficult to read. It relies on the reader knowing the meaning of the abbreviations PP.H1.abf etc. The numbers are also currently presented in scientific format, which is not helpful as, unlike p-values, posterior probabilities close to 1 (e.g. >0.9) are of interest. In general the two numbers of interest are PP.H3.abf (the posterior probability that both traits are associated, but with different causal variants) and PP.H4.abf (the posterior probability that both traits are associated and share a causal variant). If these are both close to 0, then either there's not enough power to test the colocalization (i.e. if one of PP.H1.abf or PP.H2.abf are >0.9), or there's no association (i.e. if PP.H0.abf is > 0.9, which we know is not true). When presenting the results, it's also useful to see a visual comparison of the pQTL and GWAS signals at the locus, i.e. by showing LocusZoom plots of the pQTL and GWAS signals side-by-side as a supp figure.

Reply 3.3: We appreciate Reviewer's constructive suggestion. We provided a visual comparison of the pQTL and GWAS signals at the locus for both UKB cohort and finn-b-19_HEARTFAIL cohort by showing LocusZoom plots of the pQTL and GWAS signals side-by-side (*Supplemental Figure 20*). The plots showed that the pQTL and GWAS for post-AMI HF signals were all around the S100A9 gene (*Revised position: page13, lines 12-16 in Result section*).

Supplemental Figure 20. LocusZoom plots of the pQTL and GWAS signals within 20 kb of either side of S100A9

(Top) LocusZoom plot indicating pQTL of S100A9 in serum. (Middle) LocusZoom plot indication GWAS signals for post-AMI HF in UKB cohort. (Bottom) LocusZoom plot indication GWAS signals for general HF in finn-b-19 HEARTFAIL.

REFERENCES

1. Gerstein HC, Mann JF, Yi Q, Zinman B, Dinneen SF, Hoogwerf B, Hallé JP, Young J, Rashkow A, Joyce C, Nawaz S and Yusuf S. Albuminuria and risk of cardiovascular events, death, and heart failure in diabetic and nondiabetic individuals. *Jama*. 2001;286:421-6.
2. Razaghizad A, Sharma A, Ni J, Ferreira JP, White WB, Mehta CR, Bakris GL and Zannad F. External validation and extension of the TIMI risk score for heart failure in diabetes for patients with recent acute coronary syndrome: An analysis of the EXAMINE trial. *Diabetes, obesity & metabolism*. 2023;25:229-237.
3. Ruparelia N, Chai JT, Fisher EA and Choudhury RP. Inflammatory processes in cardiovascular disease: a route to targeted therapies. *Nat Rev Cardiol*. 2017;14:133-144.
4. Szeto HH. First-in-class cardioprotective compound as a therapeutic agent to restore mitochondrial bioenergetics. *Br J Pharmacol*. 2014;171:2029-50.

REVIEWER COMMENTS

Reviewer #3 (Remarks to the Author):

Suppl. Table 5: I still find it hard to believe that out of 139 women with a mean age of 49 years, not a single one has ever smoked or is currently smoking (0.0%), whereas 92.9% of men of similar age have smoked or are currently smoking.

I still would like to learn how many of the “AMI patients” in this report had STEMIs vs. NSTEMIs. This critical information is provided in all studies on patients with AMI.

Reviewer #4 (Remarks to the Author):

The authors have adequately responded to my previous comments. I only have minor comments remaining

Thank you for providing clarification in your response and in the corresponding caption for Supplementary Figure 16 on the meaning of the values in cells in the heatmap. Two minor points that need fixing for the final submission: (1) many of the red cells now appear to be empty, please make sure the text appears correctly in the final image upload, and (2) the r-squared values should be given as decimals for clarity, e.g. 0.84 instead of 84, 0.04 instead of 4, and so on.

There are several elements of Figure 3 that would benefit from further clarification in the caption. Please add a note in the caption as to meaning of the vertical black dashed line – is this the median follow-up time in each cohort? Is the maximum follow-up time 5 years for all three cohorts? If not please add a note to the caption.

Table S5: after reading reviewer 3’s comments the author responses, I agree with reviewer 3 that there needs to be some explanation either in the table caption or methods as to the difference in numbers of former/current smoking status between males and females, considering nearly all the males were current or former smokers, whilst none of the females were.

Lines 141-162, Supplementary Figs 5-8: Building on reviewer 3’s concerns, this whole section of results appears to be based on data from only 2 samples. Consequently, these results seem highly speculative and unreliable: I would encourage cutting this section entirely as it detracts from the overall work presented in the manuscript.

Supplemental Figure 19: axis labels should be larger for clarity. I had to zoom in quite a lot to read the plot

Lines 75-77 inappropriately overstate the consequences of associations between genetically predicted biomarker levels and outcomes. Implications of causality require these genetic predictions to meet several strict assumptions, most notably the absence of horizontal pleiotropy.

Line 264: Unclear what “(>mean+1 SD)” is meant to imply

Line 225: “Due to the limited sample size in patients with Killip class III...” – please add the sample size to the text

Line 297 “cross-validated” -> “validated”. Cross validation has specific statistical meaning which is not what is being referred to here, so using “cross-validated” could be confusing or misleading.

I have a number of further minor editorial suggestions, mainly regarding fixing issues with tense and enhancing clarity in the main text:

Line 127-129 should be past tense: “have” -> “had”

Line 146: “have” -> “had”

Line 173: “the HF-S100A8/A9 association” -> “the association between S100A8/A9 and HF”

Line 178: “HF-S100A12 association was still not significant after adjustment” -> “The associated between S100A12 and HF was not significant after adjustment”

Line 182: “is” -> “was”

Line 206: “positive” -> “positives”

Line 220: “positive” -> “positives”

Line 229: “HF-S100A8/A9 associations” -> “association between S100A8/A9 and HF”

Line 230: “were still” -> “remained”

Line 252: “are” -> “were”

Line 253: “are” -> “were”

Lines 259-260: “SNP-S100A8/A9 estimates” -> “estimated effect of the SNPs on S100A8/A9 levels”

Reviewer #3 (Remarks to the Author):

1, Suppl. Table 5: I still find it hard to believe that out of 139 women with a mean age of 49 years, not a single one has ever smoked or is currently smoking (0.0%), whereas 92.9% of men of similar age have smoked or are currently smoking.

Reply 1: We greatly appreciate the reviewer's comments. We re-surveyed the smoking status of the 588 healthy subjects through telephone inquiries. We define current smokers as those who have smoked 100 cigarettes in their lifetime and had smoked cigarettes in the past 30 days. The prevalence current smoking was 50.1% (225/449) among men and 0% (0/139) among women. The data from 5 national representative cross-sectional population-based surveys in 31 provinces in mainland China during 2007 to 2018, reported that smoking prevalence was very low at around 2% among woman. (*Revised position: page4, line13-14 in Supplemental material*)

Supplemental Table 5. Characteristics of the HCs (n = 588)

Variable	HCs (n = 588)	
	Female (n = 139)	Male (n = 449)
Current smokers	0 (0.0%)	225 (50.1%)

Current smokers refer to those who have smoked 100 cigarettes in their lifetime and had smoked cigarettes in the past 30 days.

2, I still would like to learn how many of the “AMI patients” in this report had STEMIs vs. NSTEMIs. This critical information is provided in all studies on patients with AMI.

Reply 2: We greatly appreciate the reviewer's suggestion and provide this information. The proportions of STEMIs vs. NSTEMIs were 75% vs. 25% in discovery cohort and 61.6% vs. 38.4% in validation cohort. (*Revised position: page4, line22-24 in Result section*)

Supplemental Table 4. Comparison of baseline information between the discovery cohort and the validation cohort

Variables	Discovery cohort (n=1062)	Validation cohort (n=1043)	P-value
STEMI	796 (75.0%)	642 (61.6%)	<0.001

Reviewer #4 (Remarks to the Author):

The authors have adequately responded to my previous comments. I only have minor comments remaining

Reply: We greatly appreciate the Reviewer's constructive suggestions that aim to improve the quality of our study. We have specifically addressed each concern below.

1, Thank you for providing clarification in your response and in the corresponding caption for Supplementary Figure 16 on the meaning of the values in cells in the heatmap. Two minor points that need fixing for the final submission: (1) many of the red cells now appear to be empty, please make sure the text appears correctly in the final image upload, and (2) the r-squared values should be given as decimals for clarity, e.g. 0.84 instead of 84, 0.04 instead of 4, and so on.

Reply 1: We did not describe it clearly. The red cells without numbers mean the $R^2 = 1$. We revised the R^2 values to be decimals. (*Revised position: page49, line6 in Supplemental material*)

Revised Supplemental Figure 11: S100A8/A9 genetic score construction

(A) MR study design. (B) Twenty cis-pQTLs selecting within 20 kb of either side of S100A9 and their linkage disequilibrium relationship calculated by Haploview software using CHB data as reference. The number in each cell is R^2 of its corresponding two SNPs, which is a parameter indicating the degree of linkage disequilibrium between two SNPs. The larger R^2 , the stronger the degree of linkage disequilibrium. The red cells without number mean the $R^2 = 1$.

2, There are several elements of Figure 3 that would benefit from further clarification in the caption. Please add a note in the caption as to meaning of the vertical black dashed line – is this the median follow-up time in each cohort? Is the maximum follow-up time 5 years for all three cohorts? If not please add a note to the caption.

Reply 2: The vertical black dashed line means the median follow-up time in each cohort. The maximum follow-up time is 5.8 years and 5.3 years for discovery and validation cohorts respectively. We further clarified these elements in the caption of Figure 3.

Revised Figure 3: Incidence of HF events according to S100A8/A9 levels and S100A8/A9 genetic score

Kaplan–Meier curves illustrate the timing of heart failure (HF) events in the four strata of S100A8/A9 levels and S100A8/A9 genetic score. The quartile of S100A8/A9 (ng/ml) was used to classify patients into four risk categories, including low, low-intermediate, intermediate-high, and high-risk in the discovery cohort (A) and validation cohort (B). Similarly, the quartile of the S100A8/A9 genetic score was used to classify patients in the validation cohort into four risk categories (C). The vertical black dashed line means the median follow-up time (4.2 and 2.9 years for discovery and validation cohorts respectively). The maximum follow-up time is 5.8 years and 5.3 years for discovery and validation cohorts respectively.

3, Table S5: after reading reviewer 3's comments the author responses, I agree

with reviewer 3 that there needs to be some explanation either in the table caption or methods as to the difference in numbers of former/current smoking status between males and females, considering nearly all the males were current or former smokers, whilst none of the females were.

Reply 3: We re-surveyed the smoking status of the 588 healthy subjects through telephone inquiries. We define current smokers as those who have smoked 100 cigarettes in their lifetime and had smoked cigarettes in the past 30 days. The prevalence current smoking was 50.1% (225/449) among men and 0% (0/139) among women. We revised the Supplemental Table 5 based on the new survey results. (*Revised position: page4, line13-14 in Supplemental material*)

Supplemental Table 5. Characteristics of the HCs (n = 588)

Variable	HCs (n = 588)	
	Female (n = 139)	Male (n = 449)
Current smokers	0 (0.0%)	225 (50.1%)

Current smokers refer to those who have smoked 100 cigarettes in their lifetime and had smoked cigarettes in the past 30 days.

4, Lines 141-162, Supplementary Figs 5-8: Building on reviewer 3’s concerns, this whole section of results appears to be based on data from only 2 samples. Consequently, these results seem highly speculative and unreliable: I would encourage cutting this section entirely as it detracts from the overall work presented in the manuscript.

Reply 4: As Reviewer suggested, we deleted Supplementary Figures 5-9 entirely. Figure 1, Supplemental Table 1, and Supplemental Figure 2 have also been revised accordingly.

5, Supplemental Figure 19: axis labels should be larger for clarity. I had to zoom in quite a lot to read the plot

Reply 5: We have enlarged the axis labels.

Revised Supplemental Figure 14: Dose-response curves showing the causal effect of S100A8/A9 levels on post-MI/general HF

6, Lines 75-77 inappropriately overstate the consequences of associations between genetically predicted biomarker levels and outcomes. Implications of causality

require these genetic predictions to meet several strict assumptions, most notably the absence of horizontal pleiotropy.

Reply 6: We have revised this to be “If genetically predicted portion of biomarker is associated with the outcome and meets several strict assumptions, the measured marker might have a causal effect on outcomes.” (*Revised position: page3, line24 and page4 line1-2 in Introduction section*)

7, Line 264: Unclear what “(>mean+1 SD)” is meant to imply

Reply 7: We have revised this to be “Among these 588 HCs, the increase in the S100A8/A9 genetic score was significantly associated with the high-risk S100A8/A9 levels (refer to S100A8/A9 levels are greater than the mean plus one standard deviation of this cohort) (odds ratio [OR] per SD: 1.40 [95% CI: 1.11–1.76], P = 0.004). (*Revised position: page10, line20-23 in Result section*)

8, Line 225: “Due to the limited sample size in patients with Killip class III...” – please add the sample size to the text

Reply 8: We have revised this to be “Due to the limited sample size in patients with Killip class III (20 and 14 patients in discovery and validation cohort), a rising trend was only observed between Killip class III and class II.” (*Revised position: page9, line6-8 in Result section*)

9, Line 297 “cross-validated” -> “validated”. Cross validation has specific statistical meaning which is not what is being referred to here, so using “cross-validated” could be confusing or misleading.

Reply 9: We have revised this as Reviewer suggested. (*Revised position: page12, line6 in Result section*)

I have a number of further minor editorial suggestions, mainly regarding fixing issues with tense and enhancing clarity in the main text:

10, Line 127-129 should be past tense: “have” -> “had”

Reply 10: We have revised into “108 proteins had higher levels”. (*Revised position: page6, line3 in Result section*)

11, Line 146: “have” -> “had”

Reply 11: We have deleted the lines 144-162.

12, Line 173: “the HF-S100A8/A9 association” -> “the association between S100A8/A9 and HF”

Reply 12: We have revised into “the association between S100A8/A9 and HF remained”. (*Revised position: page7, line3 in Result section*)

13, Line 178: “HF-S100A12 association was still not significant after adjustment” -> “The associated between S100A12 and HF was not significant after adjustment”

Reply 13: We have revised into “The associated between S100A12 and HF was not significant after adjustment”. (*Revised position: page7, line7-8 in Result section*)

14, Line 182: “is” -> “was”

Reply 14: We have revised into “death not caused by HF was a competing risk factor”. (*Revised position: page7, line13 in Result section*)

15, Line 206: “positive” -> “positives”

Reply 15: We have revised into “additional false positives”. (*Revised position: page8, line11 in Result section*)

16, Line 220: “positive” -> “positives”

Reply 16: We have revised into “additional false positives”. (*Revised position: page9, line1 in Result section*)

17, Line 229: “HF-S100A8/A9 associations” -> “association between S100A8/A9 and HF”

Reply 17: We have revised into “association between S100A8/A9 and HF”. (*Revised position: page9, line11-12 in Result section*)

18, Line 230: “were still” -> “remained”

Reply 18: We have revised into “remained independently associated”. (*Revised position: page9, line13 in Result section*)

19, Line 252: “are” -> “were”

Reply 19: We have revised into “were associated”. (*Revised position: page10, line9 in Result section*)

20, Line 253: “are” -> “were”

Reply 20: We have revised into “were associated”. (*Revised position: page10, line10 in Result section*)

21, Lines 259-260: “SNP-S100A8/A9 estimates” -> “estimated effect of the SNPs on S100A8/A9 levels”

Reply 21: We have revised into “estimated effect of the SNPs on S100A8/A9 levels”. (*Revised position: page10, line17 in Result section*)